# Decision Making under the Exponential Family:
# Distributionally Robust Optimisation with Bayesian Ambiguity Sets

**Charita Dellaporta** [* 1 2]  **Patrick O'Hara** [* 3]  **Theodoros Damoulas** [2 3 4]

## Abstract

Decision making under uncertainty is challenging as the data-generating process (DGP) is often unknown. Bayesian inference proceeds by estimating the DGP through posterior beliefs on the model's parameters. However, minimising the expected risk under these beliefs can lead to suboptimal decisions due to model uncertainty or limited, noisy observations. To address this, we introduce Distributionally Robust Optimisation with Bayesian Ambiguity Sets (DRO-BAS) which hedges against model uncertainty by optimising the worst-case risk over a posterior-informed ambiguity set. We provide two such sets, based on the posterior expectation (DRO-BAS$_{PE}$) or the posterior predictive (DRO-BAS$_{PP}$) and prove that both admit, under conditions, strong dual formulations leading to efficient single-stage stochastic programs which are solved with a sample average approximation. For DRO-BAS$_{PE}$, this covers all conjugate exponential family members while for DRO-BAS$_{PP}$ this is shown under conditions on the predictive's moment generating function. Our DRO-BAS formulations outperform existing Bayesian DRO on the Newsvendor problem and achieve faster solve times with comparable robustness on the Portfolio problem.

## 1. Introduction

Decision-makers regularly attempt to optimise an objective under uncertainty with only finite sampling access – via independently and identically distributed (i.i.d.) observations –

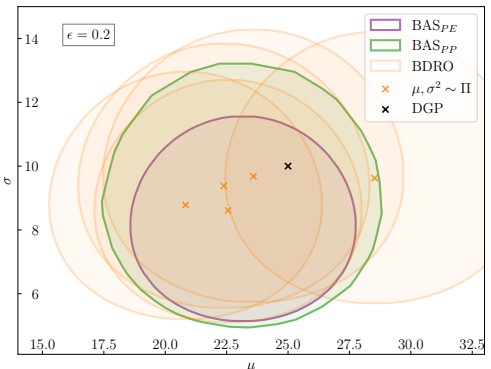

*Figure 1.* Visualisation of our BAS$_{PE}$ and BAS$_{PP}$ ambiguity sets versus BDRO (Shapiro et al., 2023) for Normal likelihood with Normal-Gamma posterior. BAS$_{PE}$ contains distributions with expected (under the posterior $\Pi$) KL divergence from the model of at most $\epsilon$. BAS$_{PP}$ contains distributions with KL divergence at most $\epsilon$ from the posterior predictive distribution. BDRO minimises the *expected* worst-case risk between each KL-based ambiguity set based on posterior samples $\mu, \sigma^2 \sim \Pi$ (orange crosses).

from the data-generating process (DGP). Given the observations, parametric model-based inference estimates the DGP with a distribution $\mathbb{P}_\theta$ indexed by parameters $\theta \in \Theta$ in parameter space $\Theta \subseteq \mathbb{R}^k$. In a Bayesian framework, the data is combined with a prior to obtain posterior beliefs about $\theta$ and the decision maker can now minimise the risk under the Bayesian estimator.

However, the estimator is likely different from the true DGP due to model and data uncertainty: the number of observations may be small; the data may be noisy; and the prior or model may be misspecified. Minimising the risk under the Bayesian model inherits any estimation error, and leads to overly optimistic decisions on out-of-sample scenarios even if the estimator is unbiased: this phenomenon is called the optimiser's curse (Kuhn et al., 2019). For example, if the number of observations is small and the prior is overly concentrated, the decision will likely be overly optimistic.

To hedge against the uncertainty of the estimated distribution, the field of Distributionally Robust Optimisation (DRO) minimises the expected objective function under the worst-case distribution that lies in an ambiguity set.

---

[*]Equal contribution  [1]Department of Statistics, University College London, UK [2]Department of Statistics, University of Warwick, Coventry, UK [3]Department of Computer Science, University of Warwick, Coventry, UK [4]The Alan Turing Institute, London, UK. Correspondence to: Charita Dellaporta <h.dellaporta@ucl.ac.uk>, Patrick O'Hara <Patrick.H.O-Hara@warwick.ac.uk>.

*Proceedings of the $42^{nd}$ International Conference on Machine Learning*, Vancouver, Canada. PMLR 267, 2025. Copyright 2025 by the author(s).

Discrepancy-based ambiguity sets contain distributions that are close to a nominal one in the sense of some discrepancy measure such as the Kullback-Leibler (KL) divergence (Hu & Hong, 2013), Wasserstein distance (Kuhn et al., 2019) or Maximum Mean Discrepancy (Staib & Jegelka, 2019). For example, model-based methods[1] (Iyengar et al., 2023; Michel et al., 2021; 2022) consider a family of parametric models and create discrepancy-based ambiguity sets centred on the fitted model. However, these works do not capture uncertainty about the parameters which can lead to a nominal distribution far away from the DGP when the data is limited. The principled framework for capturing such uncertainty is Bayesian inference. Existing DRO methodologies like Bayesian DRO (BDRO) introduced by Shapiro et al. (2023) that are informed by parameter posterior beliefs, do not correspond to a worst-case risk objective and hence do not give rise to a single, interpretable worst-case risk distribution (see Figure 1).

We introduce DRO with Bayesian Ambiguity Sets (DRO-BAS): a framework for robust decision-making under uncertainty based on *posterior-informed ambiguity sets*. Our contributions are:

1. We define Bayesian Ambiguity Sets (BAS) which leverage posterior beliefs about the parameters of interest. We provide two distinct formulations of BAS: one containing distributions with small KL divergence from the posterior predictive distribution (DRO-BAS$_{\text{PP}}$) and another one including those with small expected KL divergence from the candidate model (DRO-BAS$_{\text{PE}}$). Similarly to discrepancy-based DRO, the decision maker solves a single worst-case risk minimisation problem.

2. We show that DRO-BAS$_{\text{PP}}$ attains a dual formulation which is an efficient single-stage stochastic program. Additionally, for models within the conjugate exponential family, we demonstrate that the dual program of DRO-BAS$_{\text{PE}}$ is also single-stage and can be solved exactly when a linear objective function and a Gaussian likelihood are used. When the objective function is non-linear but convex, the duals can be solved with a sample average approximation (SAA) (also known as Monte Carlo). Finally, we provide the worst-case risk distribution form as well as finite-sample results on the optimal tolerance level.

3. On the Newsvendor problem, we show that DRO-BAS Pareto dominates existing Bayesian DRO formulations (Shapiro et al., 2023) when evaluating the out-of-sample mean-variance trade-off of the objective function when the number of SAA samples is small. On the real-world Portfolio problem, DRO-BAS solves

instances faster than Bayesian DRO with comparable out-of-sample robustness.

## 2. Background

Let $x \in \mathcal{X}$ be a decision-making variable that is chosen to minimise a stochastic objective function $f : \mathcal{X} \times \Xi \to \mathbb{R}$, where $\mathcal{X} \subseteq \mathbb{R}^D$ is the set of feasible decisions and $\Xi \subseteq \mathbb{R}^D$ is the data space[2]. Let $\mathbb{P}^\star \in \mathcal{P}(\Xi)$ be the data-generating process (DGP) where $\mathcal{P}(\Xi)$ is the space of Borel distributions over $\Xi$. We are given $n$ i.i.d. observations $\mathcal{D} \triangleq \xi_{1:n} \sim \mathbb{P}^\star$. Model-based inference considers a family of models $\mathcal{P}_\Theta \triangleq \{\mathbb{P}_\theta : \theta \in \Theta\} \subset \mathcal{P}(\Xi)$ where each $\mathbb{P}_\theta$ has probability density function $p(\xi|\theta)$ for parameter space $\Theta \subseteq \mathbb{R}^k$. In a Bayesian framework, data $\mathcal{D}$ is combined with a prior $\pi(\theta)$ to obtain posterior beliefs about $\theta$ through $\Pi(\theta|\mathcal{D})$. Under expected-value risk, Bayesian Risk Optimisation (Wu et al., 2018) solves a stochastic optimisation problem:

$$(BRO) \qquad \min_{x \in \mathcal{X}} \mathbb{E}_{\theta \sim \Pi(\theta|\mathcal{D})} \left[ \mathbb{E}_{\xi \sim \mathbb{P}_\theta}[f_x(\xi)] \right]. \qquad (1)$$

where $f_x(\xi) \triangleq f(x, \xi)$ is the objective function. However, decision-makers sometimes seek worst-case protection. The closest work to ours, using parametric Bayesian inference to inform the optimisation problem, is Bayesian DRO (BDRO) by Shapiro et al. (2023). BDRO takes an *expected worst-case* approach under the posterior distribution. More specifically, let $\mathcal{B}_\epsilon(\mathbb{P}_\theta) \triangleq \{\mathbb{Q} \in \mathcal{P}(\Xi) : d_{\text{KL}}(\mathbb{Q}\|\mathbb{P}_\theta) \leq \epsilon\}$ be the KL discrepancy-based ambiguity set centered on distribution $\mathbb{P}_\theta$, where $d_{\text{KL}}$ is the KL-divergence and parameter $\epsilon \in [0, \infty)$ controls the size of the ambiguity set. Under the expected value of the posterior, BDRO solves:

$$(BDRO) \min_{x \in \mathcal{X}} \mathbb{E}_{\theta \sim \Pi(\theta|\mathcal{D})} \left[ \sup_{\mathbb{Q} \in \mathcal{B}_\epsilon(\mathbb{P}_\theta)} \mathbb{E}_{\xi \sim \mathbb{Q}}[f_x(\xi)] \right], \qquad (2)$$

where $\mathbb{E}_{\theta \sim \Pi(\theta|\mathcal{D})}[Y] \triangleq \int_\Theta Y(\theta)\Pi(\theta \mid \mathcal{D}) \, d\theta$ denotes the expectation of random variable $Y : \Theta \to \mathbb{R}$ with respect to $\Pi(\theta \mid \mathcal{D})$. A decision maker is often interested in protecting against and quantifying the worst-case risk, but BDRO does not correspond to a worst-case risk analysis. Moreover, the BDRO dual problem is a two-stage stochastic problem that involves a double expectation over the posterior and likelihood. To get a good approximation of the dual problem using SAA, a large number of samples are required, which increases the solving time of the dual problem.

Gupta (2019) introduced Near-Optimal Ambiguity Sets, based on Bayesian guarantees. This method aims to find the smallest convex ambiguity set that satisfies a *Bayesian Posterior Feasibility* guarantee. The authors also provide

---

[1]For a discussion of empirical DRO vs model-based DRO methods, see Appendix E.3.

[2]For convenience, we assume the decision variable $x \in \mathcal{X}$ and the random variable $\xi \in \Xi$ both have dimension $D$, but our work is not restricted to this case.

an upper bound on the posterior value-at-risk, constructing the ambiguity set according to the framework of Bertsimas et al. (2018). In this approach, the ambiguity set is chosen by searching over a space of potential sets and selecting the one that meets a predefined guarantee based on the posterior distribution. In contrast to BDRO, the ambiguity set is confined to members of the model family, and posterior beliefs are *not* used to inform the ambiguity set itself, but rather to guide the search for feasible sets.

Alternative DRO formulations often incorporate nonparametric Bayesian methods. Wang et al. (2023) suggested using a nonparametric prior over the DGP $\mathbb{P}^\star$ to induce distributional uncertainty. This nonparametric prior is chosen to be a Dirichlet Process (DP) prior with prior centring measure $\mathbb{F} \in \mathcal{P}(\Xi)$. The authors suggest inducing distributional robustness by minimising the worst-case loss over an ambiguity set for $\mathbb{F}$. Bariletto & Ho (2024) also address DGP uncertainty by combining DP-based posterior beliefs about the DGP with a decision-theoretic framework that models smooth ambiguity-averse preferences, arising from the economic decision theory literature (Cerreia-Vioglio et al., 2011). This framework was recently generalised by Bariletto et al. (2024) to handle heterogeneous data sources through hierarchical DPs. Unlike the previously mentioned BDRO, these methods are not well-suited for situations where the decision-maker has a parametric model and seeks to incorporate parameter posterior beliefs.

## 3. Bayesian Ambiguity Sets

The goal of this work is to formulate a family of optimisation problems that, based on parameter posterior beliefs, produce decisions with worst-case risk protection. We introduce *Bayesian Ambiguity Sets (BAS)* which achieve this through two ambiguity set constructions. We first propose a BAS based on the posterior predictive (BAS_PP) which forms a distributionally robust counterpart to the BRO objective in (1). To overcome some of its drawbacks, we further propose BAS based on a posterior expectation (BAS_PE). The difference between the two lies in the position of the posterior expectation in the constraint. The two ambiguity sets, for tolerance level $\epsilon$, consider probability measures $\mathbb{Q} \in \mathcal{P}(\Xi)$ with density function $q(\xi)$ such that:

$$(BAS_{PP}) \quad d_{\mathrm{KL}}\left(q(\xi), \int_\Theta p(\xi \mid \theta)d\Pi(\theta \mid \mathcal{D})\right) \leq \epsilon,$$

$$(BAS_{PE}) \quad \int_\Theta d_{\mathrm{KL}}\left(q(\xi), p(\xi \mid \theta)\right) d\Pi(\theta \mid \mathcal{D}) \leq \epsilon.$$

### 3.1. BAS via Posterior Predictive (BAS_PP)

Given posterior $\Pi$ and risk tolerance level $\epsilon \in [0, \infty)$ we extend the Bayesian risk objective in (1) to a worst-case

risk minimisation, controlled by $\epsilon$. Expected-value risk in (1) is equivalent to minimising the expected risk under the posterior predictive $\mathbb{P}_n \in \mathcal{P}(\Xi)$ that has probability density function defined by:

$$p_n(\xi^\star \mid \mathcal{D}) \triangleq \int_\Theta p(\xi^\star \mid \theta)d\Pi(\theta \mid \mathcal{D}). \tag{3}$$

First we propose *Bayesian Ambiguity Sets with the posterior predictive (BAS_PP)*:

$$\mathcal{B}_\epsilon(\mathbb{P}_n) \triangleq \{\mathbb{Q} \in \mathcal{P}(\Xi) : d_{\mathrm{KL}}(\mathbb{Q}, \mathbb{P}_n) \leq \epsilon\} \tag{4}$$

where $\mathbb{Q} \in \mathcal{P}(\Xi)$ is a distribution in the ambiguity set and $d_{\mathrm{KL}} : \mathcal{P}(\Xi) \times \mathcal{P}(\Xi) \to \mathbb{R}$ is the *KL divergence*. The ambiguity set is informed by the posterior predictive $\mathbb{P}_n$: it consists of all probability measures $\mathbb{Q} \in \mathcal{P}(\Xi)$ which are absolutely continuous with respect to $\mathbb{P}_n$, i.e. $\mathbb{Q} \ll \mathbb{P}_n$, and have *KL-divergence* to $\mathbb{P}_n$ less than or equal to $\epsilon$. Thus, $\epsilon$ dictates the desired amount of risk in the decision. For a given decision $x$, the *worst-case risk* is

$$\mathcal{R}_{\mathcal{B}_\epsilon(\mathbb{P}_n)}(f_x) \triangleq \sup_{\mathbb{Q} \in \mathcal{B}_\epsilon(\mathbb{P}_n)} \mathbb{E}_{\xi \sim \mathbb{Q}} \ [f_x(\xi)]. \tag{5}$$

The posterior $\Pi(\theta \mid \mathcal{D})$ targets the KL minimiser between the model family and $\mathbb{P}^\star$ (Walker, 2013), hence it is natural to choose the *KL divergence* in (4). This is because as $n \to \infty$ the posterior collapses to $\theta^\star \triangleq \arg\min_{\theta \in \Theta} d_{\mathrm{KL}}(\mathbb{P}^\star, \mathbb{P}_\theta)$ and the posterior predictive is equal to $p_n(\xi^\star \mid \theta^\star)$. Thus, for a well-specified model family, which occurs when $\mathbb{P}^\star \equiv \mathbb{P}_{\theta^\star} \in \mathcal{P}_\Theta$, we have that in the limit of infinite observations, $\mathcal{B}_\epsilon(\mathbb{P}_n)$ is a KL-based ambiguity set with $\mathbb{P}^\star$ the nominal distribution. The goal of the decision maker is to choose $x$ that minimises the worst-case risk, yielding the problem:

$$(DRO\text{-}BAS_{PP}) \qquad \min_x \mathcal{R}_{\mathcal{B}_\epsilon(\mathbb{P}_n)}(f_x) \tag{6}$$

Ambiguity sets of the form $\mathcal{B}_\epsilon(\mathbb{P})$ based on the KL-divergence have been previously suggested in a non-Bayesian DRO context by Hu & Hong (2013), who studied the optimisation of this problem for a general nominal distribution $\mathbb{P}$. For the dual of the worst-case risk $\mathcal{R}_{\mathcal{B}_\epsilon(\mathbb{P})}(f_x)$ to be well-defined, the following property of the nominal distribution $\mathbb{P}$ is required.

**Property 3.1** (Finite moment-generating function of $f_x$). Let $\mathbb{P} \in \mathcal{P}(\Xi)$. For all $x \in \mathcal{X}$, there exists $t > 0$ such that:

$$\mathbb{E}_{\xi \sim \mathbb{P}}\left[\exp(tf_x(\xi))\right] < \infty. \tag{7}$$

If $\mathbb{P}_n$ satisfies this property, we have the following result.

**Proposition 3.2.** *Assume $\mathbb{P}_n$ satisfies Property 3.1 for all $\mathcal{D} \subset \Xi^n$. Then, for any $\epsilon > 0$, the worst-case risk in (5) is:*

$$\mathcal{R}_{\mathcal{B}_\epsilon(\mathbb{P}_n)}(f_x) = \inf_{\gamma \geq 0} \gamma\epsilon + \gamma \ln \mathbb{E}_{p_n(\xi^\star \mid \mathcal{D})}\left[e^{\frac{f_x(\xi)}{\gamma}}\right]. \tag{8}$$

The proof can be found in Appendix B.1. This result offers *strong duality* and the resulting optimisation problem is a single-stage stochastic program. Unlike BDRO, the problem in (6) not only provides a worst-case approach but also requires sampling from only a single expectation when the posterior predictive is available.

Assuming that Property 3.1 holds in Proposition 3.2 requires the moment-generating function (MGF) of $f_x(\xi^\star)$ under the posterior predictive $\mathbb{P}_n$ to be finite. This is a standard assumption made in KL-based DRO methods (Hu & Hong, 2013; Shapiro et al., 2023) ensuring that the dual objective in (8) is well-defined. However, this assumption can be violated when e.g. the model is a Normal distribution with a conjugate Normal-Gamma prior, leading to a Student-t posterior predictive (see e.g. Murphy, 2023) which has an infinite MGF. A similar problem occurs when the Exponential distribution with a conjugate Gamma prior is used. Thus, the use of such an ambiguity set can be limiting in the model and objective function choices. A way to overcome this limitation is to approximate the primal problem in (6) using the ambiguity set $\mathcal{B}_\epsilon(\hat{\mathbb{P}}_n)$ based on a finite sample approximation $\hat{\mathbb{P}}_n$ of the posterior predictive where:

$$\xi^\star_{1:M} \stackrel{\text{iid}}{\sim} \mathbb{P}_n, \quad \hat{\mathbb{P}}_{n,M} \triangleq \frac{1}{M}\sum_{i=1}^{M}\delta_{\xi^\star_i}. \tag{9}$$

Although for finite $M$, the dual formulation can be applied with $\hat{\mathbb{P}}_{n,M}$, the resulting optimisation problem is always a SAA and does not solve the original primal in (6) to optimality. Hence, we offer an alternative BAS formulation based on a posterior expectation.

### 3.2. BAS via Posterior Expectation (BAS_PE)

Consider *Bayesian ambiguity sets with the Posterior Expectation (BAS_PE)* defined as:

$$\mathcal{A}_\epsilon(\Pi) \triangleq \{\mathbb{Q} \in \mathcal{P} : \mathbb{E}_{\theta \sim \Pi}[d_{\mathrm{KL}}(\mathbb{Q}, \mathbb{P}_\theta)] \le \epsilon\}. \tag{10}$$

The ambiguity set considers all probability measures $\mathbb{Q} \in \mathcal{P}(\Xi)$, which are absolutely continuous with respect to $\mathbb{P}_\theta$ ($\mathbb{Q} \ll \mathbb{P}_\theta$) for all $\theta \in \Theta$ and have at most $\epsilon$ *expected* (with respect to $\Pi$) *KL divergence* to $\mathbb{P}_\theta$. The shape of our ambiguity set is posterior-driven: we emphasise this point by taking the posterior $\Pi$ as an argument to the ambiguity set $\mathcal{A}_\epsilon(\Pi)$. This is contrary to standard discrepancy-based ambiguity sets $\mathcal{B}_\epsilon(\cdot)$ as in (4) which are centred on a fixed nominal distribution. We will later prove that BAS_PE can be reduced to an ambiguity set of the form $\mathcal{B}_\epsilon(\cdot)$ for exponential family models, allowing for efficient computation. For fixed decision $x$, the *worst-case risk* is:

$$\mathcal{R}_{\mathcal{A}_\epsilon(\Pi)}(f_x) \triangleq \sup_{\mathbb{Q} \in \mathcal{A}_\epsilon(\Pi)} \mathbb{E}_{\xi \sim \mathbb{Q}}[f_x(\xi)]. \tag{11}$$

Similarly to the BAS_PP worst-case risk in (5), our formulation (11) is still a worst-case approach, keeping with DRO

tradition, instead of BDRO's *expected* worst-case formulation (2). The goal is to minimise the worst-case risk:

$$(\textit{DRO-BAS}_{PE}) \qquad \min_x \ \mathcal{R}_{\mathcal{A}_\epsilon(\Pi)}(f_x). \tag{12}$$

We derive an upper bound for the worst-case risk in (11):

**Proposition 3.3.** *Assume $\mathbb{P}_\theta$ satisfies Property 3.1 for all $\theta \in \Theta$. Then the worst-case risk satisfies:*

$$\mathcal{R}_{\mathcal{A}_\epsilon(\Pi)}(f_x) \le \inf_{\gamma \ge 0} \ \gamma\epsilon + \mathbb{E}_\Pi \left[ \gamma \ln \mathbb{E}_{\mathbb{P}_\theta} \left[ e^{\frac{f_x(\xi)}{\gamma}} \right] \right]. \tag{13}$$

*Proof sketch.* We introduce a Lagrangian variable $\gamma \ge 0$ for the constraint $\mathbb{E}_{\theta \sim \Pi}[d_{\mathrm{KL}}(\mathbb{Q}, \mathbb{P}_\theta)] \le \epsilon$ from (10) and write the dual in terms of the convex conjugate $(\mathbb{E}_\Pi [\gamma d_{\mathrm{KL}}(\cdot\|\mathbb{P}_\theta)])^\star (f_x)$ of the expected KL-divergence. The upper bound follows by Jensen's inequality because the conjugate is a convex function. The result follows from the convex conjugate of the KL-divergence (Bayraksan & Love, 2015; Shapiro et al., 2023). See Appendix B.2 for a detailed and full proof. □

While Proposition 3.3 provides a weak duality upper bound applicable to general Bayesian models, its primary purpose is to motivate the need for more tractable formulations. In particular, this bound highlights the challenges of working with expected $\phi$-divergences, like the KL. To address this, Section 3.3 focuses on exponential family models, where we can move beyond the weak duality bound and derive a strong duality result in Theorem 3.6. This analysis requires the closed-form expression of the expected KL divergence (Lemma 3.5), which plays a central role in obtaining the convex conjugate and deriving novel results on tolerance levels, worst-case distributions, and tractable optimisation.

### 3.3. DRO-BAS_PE for the Exponential Family

The convex conjugate $(\mathbb{E}_\Pi [\gamma d_{\mathrm{KL}}(\cdot\|\mathbb{P}_\theta)])^\star$ of the expected KL-divergence in the proof of Proposition 3.3 is not easy to obtain for a general Bayesian model, but if we can find an exact expression for this convex conjugate for specific models, then the worst-case risk (11) can be computed exactly. In this section, we derive such an expression for exponential family models with conjugate priors. When the likelihood distribution is a member of the exponential family, there exists a conjugate prior that also belongs to the exponential family (see e.g. Diaconis & Ylvisaker, 1979). Exponential families include many of the most widely used probability distributions, making them versatile for modelling across diverse settings. Additionally, they are highly appealing due to their extensive theoretical foundation and a range of valuable properties (see Diaconis & Ylvisaker, 1979; Gutiérrez-Peña et al., 1997; Gutierrez-Pena, 1997, for a detailed overview.). We use the conjugate exponential family

and the convexity of the log-partition function to derive the strong dual of the worst-case risk problem in (11). The following definition is from Murphy (2023).

**Definition 3.4** (Exponential family with conjugate prior). Let $p(\mathcal{D} \mid \eta)$ be an exponential family likelihood with natural parameter $\eta \in \Omega \subseteq \mathbb{R}^K$, scaling constant $h : \Xi \to \mathbb{R}$, log-partition function $A : \Omega \to \mathbb{R}$ such that $\Omega \triangleq \{\eta \in \mathbb{R}^K : A(\eta) < \infty\}$, and sufficient statistic $s : \Xi \to \mathbb{R}^K$ with likelihood function:

$$p(\mathcal{D} \mid \eta) = h(\mathcal{D}) \exp\left(\eta^\top s(\mathcal{D}) - n A(\eta)\right)$$

where $h(\mathcal{D}) = \prod_{i=1}^n h(\xi_i)$ and $s(\mathcal{D}) = \sum_{i=1}^n s(\xi_i)$. We consider the prior distribution:

$$\pi(\eta \mid \breve{\tau}, \breve{\nu}) = \frac{1}{Z(\breve{\tau}, \breve{\nu})} \exp\left(\breve{\tau}^\top \eta - \breve{\nu} A(\eta)\right) \qquad (14)$$

where $\breve{\tau}, \breve{\nu}$ are prior hyperparameters and $Z(\breve{\tau}, \breve{\nu})$ is the normalising constant. The conjugate posterior $\Pi(\eta \mid \widehat{\tau}, \widehat{\nu})$ has the same form as (14) with parameters $\widehat{\tau} = \breve{\tau} + s(\mathcal{D})$ and $\widehat{\nu} = \breve{\nu} + n$.

We start with a Lemma before presenting the main result. The proof can be found in Appendix B.3.

**Lemma 3.5.** *Let $p(\xi \mid \eta)$ be an exponential family likelihood function with natural parameter $\eta$ and $\pi(\eta \mid \breve{\tau}, \breve{\nu})$, $\Pi(\eta \mid \widehat{\tau}, \widehat{\nu})$ a conjugate prior-posterior pair as in Definition 3.4. Let $\hat{\eta} \triangleq \mathbb{E}_{\Pi(\eta\mid\widehat{\tau},\widehat{\nu})}[\eta]$ and define $G : T \to \mathbb{R}$ as:*

$$G(\widehat{\tau}, \widehat{\nu}) \triangleq \mathbb{E}_{\Pi(\eta\mid\widehat{\tau},\widehat{\nu})}[A(\eta)] - A(\hat{\eta})$$

*for hyperparameter space $T$. If for almost all $\eta \in \Omega$ the partial derivatives $\partial_{\widehat{\nu}}$ and $\nabla_{\widehat{\tau}}$ of the posterior p.d.f. $\Pi : \Omega \times T \to \mathbb{R}_{\geq 0}$ exist for all $(\widehat{\nu}, \widehat{\tau}) \in T$ then:*

$$\hat{\eta} = -\nabla_{\widehat{\tau}}\left(-\ln Z(\widehat{\tau}, \widehat{\nu})\right), \qquad (15)$$

$$G(\widehat{\tau}, \widehat{\nu}) = \frac{\partial}{\partial_{\widehat{\nu}}}\left(-\ln Z(\widehat{\tau}, \widehat{\nu})\right) - A(\hat{\eta}) \geq 0 \qquad (16)$$

*and the expected KL-divergence under the posterior is:*

$$\mathbb{E}_{\Pi(\eta\mid\widehat{\tau},\widehat{\nu})}[d_{KL}(\mathbb{Q}\|\mathbb{P}_\eta)] = d_{KL}(\mathbb{Q}, \mathbb{P}_{\hat{\eta}}) + G(\widehat{\tau}, \widehat{\nu}). \qquad (17)$$

It is straightforward to establish the minimum tolerance level $\epsilon_{\min}$ required to obtain a non-empty BAS$_{\text{PE}}$. Since the KL divergence is non-negative, under the condition of Lemma 3.5, for any $\mathbb{Q} \in \mathcal{P}(\Xi)$:

$$\mathbb{E}_{\eta\sim\Pi}[d_{\text{KL}}(\mathbb{Q}\|\mathbb{P}_\eta)] = d_{\text{KL}}(\mathbb{Q}, \mathbb{P}_{\hat{\eta}}) + G(\widehat{\tau}, \widehat{\nu})$$
$$\geq G(\widehat{\tau}, \widehat{\nu}) \triangleq \epsilon_{\min}(n). \qquad (18)$$

We are now ready to present our main result, the proof of which can be found in Appendix B.4.

**Theorem 3.6.** *Suppose the conditions of Lemma 3.5 hold and $\epsilon \geq \epsilon_{\min}(n)$ as in (18). Furthermore, assume $p(\xi \mid \hat{\eta})$ satisfies Property 3.1 for all $\hat{\eta} \in \Omega$. Then the worst-case risk $\mathcal{R}_{\mathcal{A}_\epsilon(\Pi)}(f_x)$ in (11) is equal to:*

$$\inf_{\gamma \geq 0} \gamma(\epsilon - G(\widehat{\tau}, \widehat{\nu})) + \gamma \ln \mathbb{E}_{p(\xi\mid\hat{\eta})}\left[e^{\frac{f_x(\xi)}{\gamma}}\right]. \qquad (19)$$

Observe that this dual formulation (19) closely mirrors the dual of the problem in (8), albeit with a different tolerance level and expectation law. However, in this case, assuming Property 3.1 requires $f_x(\xi)$ to have a finite MGF with respect to a member of the model family, namely $p(\xi \mid \hat{\eta})$. Contrary to the posterior predictive case, this is satisfied for exponential family models with many objective functions such as the linear and piecewise-linear ones we use in our experiments. The same model assumption is imposed by BDRO (Shapiro et al., 2023).

Using this theorem, we can derive an immediate connection between BAS$_{\text{PE}}$ (10) and KL-based ambiguity sets of the form $\mathcal{B}_\epsilon(\cdot)$ such as BAS$_{\text{PP}}$:

**Corollary 3.7.** *Let $\epsilon' \triangleq \epsilon - G(\widehat{\tau}, \widehat{\nu})$. Then in the exponential family case, $\mathcal{A}_\epsilon(\Pi) \equiv \mathcal{B}_{\epsilon'}(P_{\hat{\eta}})$.*

Hence, in the exponential family case, the two BAS formulations are different as they correspond to KL-based ambiguity sets with *distinct* nominal distributions. Additional insights on their relationship are discussed in Appendix D.

It is worth noting that although the dual formulations of the DRO-BAS formulations and the PDRO method of Iyengar et al. (2023) with the KL divergence are both model-based, they are fundamentally different. Our ambiguity sets are a posteriori informed, integrating prior beliefs and data evidence. Consequently, the resulting nominal distribution ($\mathbb{P}_n$ for DRO-BAS$_{\text{PP}}$ and $\mathbb{P}_{\hat{\eta}}$ for DRO-BAS$_{\text{PE}}$) contains all the information from our posterior beliefs, including their uncertainty quantification, unlike the point estimator approach of PDRO. This allows us to propagate uncertainty from the posterior beliefs about the parameters to the ambiguity set.

### 3.4. Tolerance Level Selection

To guarantee that the DRO-BAS objectives upper-bound the expected risk under the DGP, the decision-maker aims to choose $\epsilon$ large enough so that $\mathbb{P}^\star$ is contained in the ambiguity sets. For BAS$_{\text{PE}}$ with the exponential family, Lemma 3.5 yields a closed-form expression for the optimal radius $\epsilon^\star_{\text{PE}}(n)$ in the exponential family case, by noting that:

$$\epsilon^\star_{\text{PE}}(n) \triangleq \mathbb{E}_{\eta\sim\Pi}[d_{\text{KL}}(\mathbb{P}^\star\|\mathbb{P}_\eta)] = d_{\text{KL}}(\mathbb{P}^\star, \mathbb{P}_{\hat{\eta}}) + G(\widehat{\tau}, \widehat{\nu}). \qquad (20)$$

If the model is well-specified, and hence $\mathbb{P}^\star$ and $\mathbb{P}_{\hat{\eta}}$ belong to the same exponential family, it is simple to obtain

$\epsilon^\star_{\mathrm{PE}}(n)$ based on the prior, posterior and true parameters (see Appendix A for examples). In practice, since the true parameters are unknown, we can empirically approximate $\epsilon^\star_{\mathrm{PE}}(n)$ from data $\mathcal{D}$. For any $\epsilon \geq \epsilon^\star_{\mathrm{PE}}(n) \geq \epsilon_{\min}(n)$, we have that $\mathbb{P}^\star \in \mathcal{A}_\epsilon(\Pi)$, thus the worst-case risk upper bounds the true risk under the DGP:

$$\mathbb{E}_{\xi \sim \mathbb{P}^\star}[f_x(\xi)] \leq \sup_{\mathbb{Q} \in \mathcal{A}_\epsilon(\Pi)} \mathbb{E}_{\xi \sim \mathbb{Q}}[f_x(\xi)]. \qquad (21)$$

For BAS$_{\mathrm{PP}}$, the inclusion of the DGP is achieved for:

$$\epsilon^\star_{\mathrm{PP}}(n) \triangleq d_{\mathrm{KL}}(\mathbb{P}^\star, \mathbb{P}_n). \qquad (22)$$

A possible approach would be to approximate this using the observed dataset and samples $(\xi^\star_1, \ldots, \xi^\star_M) \overset{\mathrm{iid}}{\sim} \mathbb{P}_n$ from the posterior predictive via $d_{\mathrm{KL}}(\frac{1}{n} \sum_{i=1}^n \delta_{\xi_i}, \frac{1}{M} \sum_{i=1}^M \delta_{\xi^\star_i})$. However, this can become computationally expensive in higher dimensions if the KL is not available in closed form. Alternatively, we can use the approximation of $\epsilon^\star_{\mathrm{PE}}(n)$, which is available in closed-form in the well-specified case, since $\epsilon^\star_{\mathrm{PP}}(n) \leq \epsilon^\star_{\mathrm{PE}}(n)$. This follows by Jensen's inequality since the KL divergence is convex in both arguments:

$$\epsilon^\star_{\mathrm{PP}}(n) = d_{\mathrm{KL}}(\mathbb{P}^\star \| \mathbb{E}_\Pi[\mathbb{P}_\theta]) \leq \mathbb{E}_\Pi[d_{\mathrm{KL}}(\mathbb{P}^\star \| \mathbb{P}_\theta)] = \epsilon^\star_{\mathrm{PE}}(n).$$

It follows that for any $\epsilon \geq \epsilon^\star_{\mathrm{PP}}(n)$, the risk under the DGP is upper-bounded by (5).

### 3.5. Computation

When the objective function $f_x(\xi)$ is convex in $x$, the dual of DRO-BAS$_{\mathrm{PP}}$ and DRO-BAS$_{\mathrm{PE}}$ is a convex optimisation problem that can be solved using off-the-shelf solvers. Both DRO-BAS duals are single-stage stochastic programs which can be solved in practice using SAA. In contrast, the BDRO dual is a two-stage stochastic program where samples must be taken from both the posterior and likelihood Shapiro et al. (2023). The size of the dual programs - in terms of the number of variables and size of the input data - is summarised for DRO-BAS and BDRO in Table 1. To give an example for DRO-BAS$_{\mathrm{PP}}$, the size of the input of is $\mathcal{O}(M_{\mathrm{PP}}D)$ because this is the size of the samples $\xi_1, \ldots, \xi_{M_{\mathrm{PP}}} \sim \mathbb{P}_n$; the number of variables for DRO-BAS$_{\mathrm{PP}}$ is $\mathcal{O}(D + M_{\mathrm{PP}})$ because there are $D$ decision variables for each dimension of $x$, $M_{\mathrm{PP}}$ epigraphical variables for each sample $\xi_i \sim \mathbb{P}_n$, and one Lagrangian variable.

Next, we consider the special case of DRO-BAS$_{\mathrm{PE}}$ when the objective is a linear function $f_x(\xi) = \xi^T x$, the likelihood is a multivariate Gaussian $\mathcal{N}(\xi \mid \mu, \Sigma)$, and the posterior is a Normal-inverse-Wishart. Posterior parameters $\hat{\mu}, \hat{\Sigma}$ can be derived from (15) for $\hat{\eta}$ (see Appendix A.3). By the MGF of a Gaussian, we have

$$\mathbb{E}_{\mathcal{N}(\xi|\hat{\mu},\hat{\Sigma})}\left[\exp\left(\frac{\xi^T x}{\gamma}\right)\right] = \exp\left(\frac{\hat{\mu}^T x}{\gamma} + \frac{x^T \hat{\Sigma} x}{2\gamma^2}\right)$$

We can solve the infimum over $\gamma$ in (19) to obtain

$$\mathcal{R}_{\mathcal{A}_\epsilon(\Pi)}(f_x) = \hat{\mu}^T x + \sqrt{2\left(\epsilon - G(\hat{\mu}, \hat{\Sigma})\right)}\sqrt{x^T \hat{\Sigma} x}.$$

Importantly, the resulting minimisation dual for DRO-BAS$_{\mathrm{PE}}$ over decision $x$ is closed form and we do not need to sample from either the posterior or the likelihood. Thus, the number of variables in the dual is $\mathcal{O}(D)$ (the dimension of $x$) and the dual input size is $\mathcal{O}(D^2)$ (the size of $\hat{\Sigma}$).

Notably, BAS$_{\mathrm{PP}}$ does not admit such a closed-form due to the absence of a finite MGF under the Student-t distribution, which in this case corresponds to the posterior predictive law. As discussed in Section 3.1, this is a limitation of DRO-BAS$_{\mathrm{PP}}$ which leads to a violation of Property 3.1 and does not allow an exact dual formulation in Proposition 3.2 for many exponential family cases.

### 3.6. Worst-case BAS Distribution

The understanding of the structure of the proposed BAS sets is facilitated through the worst-case distributions maximising the objectives in (5) and (11). Indeed, by following the argument in Hu & Hong (2013) (Eq. 8), it can be shown that, for some $\gamma^\star(x) > 0$ minimising the dual formulation in (8), the worst-case distribution maximising the DRO-BAS$_{\mathrm{PP}}$ objective in (6) has probability density equal to:

$$p^\star(\xi', \gamma^\star(x)) := \frac{p_n(\xi') \exp\left(\frac{f_x(\xi')}{\gamma^\star(x)}\right)}{\mathbb{E}_{p_n}\left[\exp\left(\frac{f_x(\xi)}{\gamma^\star(x)}\right)\right]}, \quad \xi' \in \Xi.$$

This means that the distribution attaining the worst-case risk is proportional to the posterior predictive and has the same support. Similarly, it can be shown that, in the exponential family case, the distribution that attains the worst-case risk according to the DRO-BAS$_{\mathrm{PE}}$ objective in (11) is:

$$p^\star(\xi', \gamma^\star(x)) = \frac{\tilde{p}(\xi')}{\int_\Xi \tilde{p}(\xi)d\xi}, \quad \xi' \in \Xi$$

where $\tilde{p}(\xi) = h(\xi)\exp\left(\hat{\eta}s(\xi) - A(\hat{\eta}) + f_x(\xi')/\gamma^\star(x)\right)$ and $\gamma^\star(x)$ minimises the dual problem in (19).

Note that for BDRO, a single worst-case distribution does not exist because it considers an expected worst-case approach (see (2), Figure 1) rather than a worst-case approach, as advocated by DRO methods. If we sample $\theta_j \sim \Pi$ from the posterior for BDRO, then one can obtain a worst-case distribution for each $\mathbb{P}_{\theta_j}$ via the argument in Eq. (8) of Hu & Hong (2013). However, notice that in general, the minimiser of an expected objective is not the same as the average of the minimisers of the individual objectives. Hence, even looking at the posterior mean or mode of the worst-case minimisers of the inner worst-case objectives in (2) would not necessarily give us a single distribution $p$ that yields a worst-case risk of the form $\mathbb{E}_{\xi \sim p}[f(x, \xi)]$ corresponding to the risk minimised by BDRO.

*Table 1.* The number of variables and the size of the input for two cases of the dual program in terms of the dimension $D$ (of $\xi$ and $x$) and the number of likelihood samples $M_\xi$, posterior samples $M_\theta$, and posterior predictive samples $M_{\text{PP}}$. Variables include decision, Lagrangian, and epigraphical. We note the results for DRO-BAS$_{\text{PE}}$ refer to the exponential family case.

| Algorithm | $\mathbb{P}$ | Property 3.1 holds for $(f, \mathbb{P})$ | | Linear $f$; $\mathbb{P}_\theta = \mathcal{N}(\mu, \Sigma)$ | |
| | | Number of variables | Input size | Number of variables | Input size |
| --- | --- | --- | --- | --- | --- |
| DRO-BAS$_{\text{PE}}$ | $\mathbb{P}_\theta$ | $\mathcal{O}(D + M_\xi)$ | $\mathcal{O}(M_\xi D)$ | $\mathcal{O}(D)$ | $\mathcal{O}(D^2)$ |
| DRO-BAS$_{\text{PP}}$ | $\mathbb{P}_n$ | $\mathcal{O}(D + M_{\text{PP}})$ | $\mathcal{O}(M_{\text{PP}} D)$ | $\mathcal{O}(D + M_{\text{PP}})$ | $\mathcal{O}(M_{\text{PP}} D)$ |
| BDRO | $\mathbb{P}_\theta$ | $\mathcal{O}(D + M_\theta M_\xi)$ | $\mathcal{O}(M_\theta M_\xi D)$ | $\mathcal{O}(D + M_\theta)$ | $\mathcal{O}(M_\theta D^2)$ |

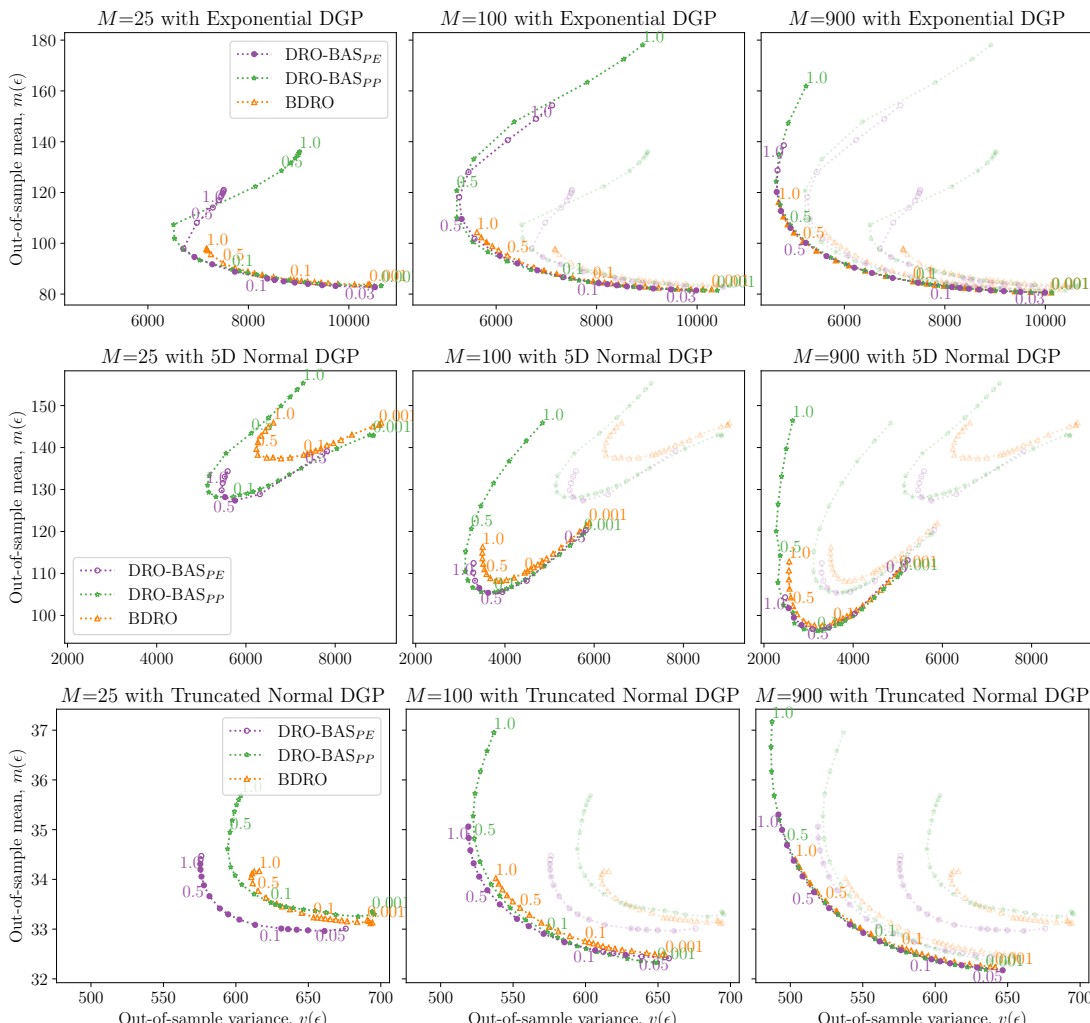

*Figure 2.* The out-of-sample mean-variance tradeoff for exponential DGP (top row), 5D multivariate normal DGP (middle row), and truncated normal DGP (bottom row). We vary $\epsilon$ when the total number of SAA samples is 25 (left), 100 (middle), and 900 (right) and plot the markers in bold. For illustration purposes, blurred markers/lines show the other smaller values of $M$; for example, the right column shows $M = 900$ in bold and $M = 25, 100$ in blurred. Each marker corresponds to a single value of $\epsilon$, some of which are labelled (e.g. $\epsilon = 0.1, 0.5, 3.0$). Filler markers indicate the given $\epsilon$ lies on the Pareto frontier.

## 4. Experiments

We evaluate DRO-BAS against BDRO on two classical problems: the Newsvendor and the Portfolio problems. We focus our analysis on Bayesian formulations of DRO, however, we discuss empirical DRO and provide additional experiments in Appendix E.3. For DRO-BAS$_{\text{PE}}$, if $\epsilon < \epsilon_{\min}(n)$, then the problem in Theorem 3.6 is unbounded and we

ignore this configuration. For DRO-BAS$_{\text{PP}}$ we use a finite sample approximation $\hat{\mathbb{P}}_{n,M}$ of the posterior predictive $\mathbb{P}_n$ as in (9). For a given $\epsilon$, we calculate the out-of-sample (OOS) mean $m(\epsilon)$ and variance $v(\epsilon)$ of the objective function $f$ across $J$ different datasets. Further details and experiments can be found in Appendix E. The code to reproduce the experiments is available at https://github.com/PatrickOHara/mis-dro-code.

### 4.1. DRO-BAS on the Newsvendor Problem

The goal of the Newsvendor problem is to choose an inventory level $x \in \mathbb{R}_{\geq 0}^D$ of $D$ perishable products with unknown customer demand $\xi \in \mathbb{R}^D$ that minimises the cost function $f_x(\xi) = h \max(\mathbf{0}, x - \xi) + b \max(\mathbf{0}, \xi - x)$, where $h, b$ are the holding cost and backorder cost per product unit respectively. Following Shapiro et al. (2023), we set $h = 3$ and $b = 8$. We run DRO-BAS$_{\text{PE}}$, DRO-BAS$_{\text{PP}}$, and BDRO across 24 different values of $\epsilon$ ranging from 0.001 to 1. For DRO-BAS$_{\text{PE}}$ and DRO-BAS$_{\text{PP}}$, we approximate the expectations over $p(\xi \mid \hat{\eta})$ and $p_n(\xi^\star \mid \mathcal{D})$ respectively with $M$ samples; for BDRO, we approximate the double expectation with $M_\theta$ posterior samples and $M_\xi$ likelihood samples for each posterior sample such that $M_\xi \times M_\theta = M$. We evaluate the methods on two well-specified settings (an exponential and a multivariate normal DGP) and one univariate misspecified setting (truncated normal DGP/normal model). For random seed $j = 1, \ldots, 500$, we sample $n = 20$ training observations from $\mathbb{P}^\star$, then sample $T = 50$ test points.

Figure 2 shows the OOS mean-variance tradeoff across all three DGPs. The middle row shows that, in the well-specified multivariate setting, both instantiations of DRO-BAS dominate BDRO in the sense that DRO-BAS forms a Pareto front for the OOS mean-variance tradeoff of the objective function $f$. That is, for any $\epsilon_1$, let $m_{\text{BDRO}}(\epsilon_1)$ and $v_{\text{BDRO}}(\epsilon_1)$ be the OOS mean and variance respectively of BDRO: then there exists $\epsilon_2$ for DRO-BAS with OOS mean $m_{\text{BAS}}(\epsilon_2)$ and variance $v_{\text{BAS}}(\epsilon_2)$ such that $m_{\text{BAS}}(\epsilon_2) < m_{\text{BDRO}}(\epsilon_1)$ and $v_{\text{BAS}}(\epsilon_2) < v_{\text{BDRO}}(\epsilon_1)$. On the top row of Figure 2 (well-specified Exponential DGP) DRO-BAS dominated BDRO for $\epsilon_{\text{BDRO}} \geq 0.5$ when $M = 25,100$, whilst all methods lie on the same Pareto front otherwise. In the misspecified case (Truncated Normal DGP with Normal likelihood), DRO-BAS$_{\text{PE}}$ Pareto dominates BDRO and DRO-BAS$_{\text{PP}}$ when $M = 25,100$ while all methods perform similarly when $M = 900$. Finally, for fixed $M$, the solve times for DRO-BAS and BDRO are broadly comparable (see Table 3 in Appendix E). However, BDRO requires more samples $M$ than DRO-BAS for good out-of-sample performance, likely because BDRO must evaluate a double expectation over the posterior and likelihood, contrary to the single expectation of DRO-BAS.

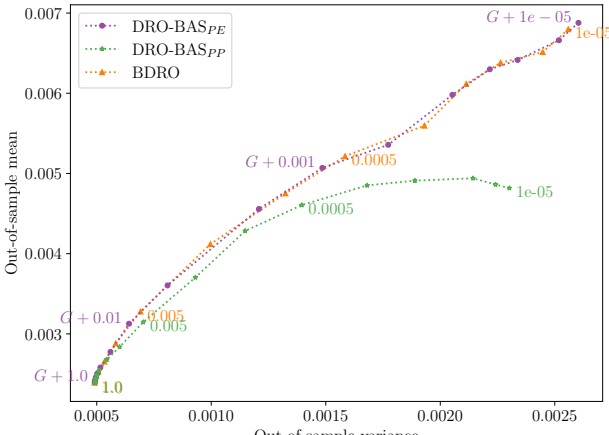

*Figure 3.* Mean-variance tradeoff of the Portfolio objective across all OOS time windows. Markers are filled similarly to Figure 2.

### 4.2. DRO-BAS on the Real-world Portfolio Problem

The goal of the Portfolio Problem is to choose the weighting of a portfolio of stocks that maximises the return. Our dataset and experimental setup follows Bruni et al. (2016) who provide weekly return data for the DowJones index on $D = 28$ stocks across 1363 weeks. Given $n = 52$ weeks of training data $\mathcal{D}_n$, we fit a model to the return $\xi$ with Normal likelihood and conjugate Normal-inverse-Wishart prior. The test dataset $\mathcal{D}_T$ contains the $T = 12$ weeks of returns immediately following the training dataset. We minimise the linear objective function $f_x(\xi) = -\xi^\top x$ (equivalent to maximising the return) such that $x_i \geq 0$ for all $i = 1, \ldots, D$ and $\sum_{i=1}^D x_i = 1$. The $J = 109$ train and test datasets are constructed using a sliding time window (see Appendix E.2).

Since the objective is linear and the likelihood is Normal, the dual objective for DRO-BAS$_{\text{PE}}$ is available in closed form (see Section 3.5). For DRO-BAS$_{\text{PP}}$, we sample from the posterior predictive $\hat{\mathbb{P}}_{n,M}$ where we set $M = 3600$ (we found DRO-BAS$_{\text{PP}}$ needs a large $M$ for good OOS performance). For BDRO, the dual program can be simplified (see Appendix C) and we need only sample $M = 900$ covariance matrices from an inverse-Wishart distribution.

Figure 3 shows that DRO-BAS$_{\text{PE}}$ and BDRO have a similar OOS mean-variance tradeoff, whilst DRO-BAS$_{\text{PP}}$ is Pareto dominated by the other two methods. Figure 7 demonstrates that the cumulative return of DRO-BAS$_{\text{PE}}$ and BDRO are broadly comparable and the returns are generally larger than DRO-BAS$_{\text{PP}}$. The discrepancy from DRO-BAS$_{\text{PE}}$ and BDRO to DRO-BAS$_{\text{PP}}$ is likely because the dual programs of DRO-BAS$_{\text{PE}}$ and BDRO can be simplified using the MGF of the Normal likelihood. The simplified dual for DRO-BAS$_{\text{PE}}$ and BDRO both yield a closed-form expression for the mean, whilst DRO-BAS$_{\text{PE}}$ also obtains the covariance

*Table 2.* Average solve times in seconds (with associated standard deviation) and sample times in seconds $\times 10^4$ on the *Portfolio Problem*. N.A. means "not applicable" due to no sampling.

| ALGORITHM | SOLVE TIME | SAMPLE ($\times 10^4$) |
|---|---|---|
| DRO-BAS$_{PE}$ | **0.01** (0.01) | N.A. |
| DRO-BAS$_{PP}$ | 1.34 (0.22) | 3.48 (0.91) |
| BDRO | 4.47 (0.62) | 44.45 (3.11) |

in closed form but BDRO must sample the covariance from the Inverse-Wishart (see Appendix C). In comparison, DRO-BAS$_{PP}$ samples from the posterior predictive without any closed-form expression for the mean or covariance.

Table 2 shows that the solve and sample time of DRO-BAS is orders of magnitude faster than BDRO on the Portfolio problem. The average solve time of DRO-BAS$_{PE}$ and DRO-BAS$_{PP}$ is 0.012 and 0.213 seconds respectively, whilst BDRO takes 3.381 seconds. Similarly, the average time taken for BDRO to sample posterior covariance matrices takes 100 times more than for DRO-BAS$_{PE}$ to calculate the parameters $\hat{\mu}$ and $\hat{\Sigma}$, and 28 times longer than DRO-BAS$_{PP}$ to sample from a Student-t distribution. Given that Figure 3 shows comparable OOS mean-variance tradeoff and cumulative returns between DRO-BAS$_{PE}$ and BDRO, we conclude that DRO-BAS$_{PE}$ is the more suitable method for achieving low OOS mean-variance in this example due to its elimination of sampling and faster solve time.

## 5. Discussion

We proposed an approach to decision-making under uncertainty through two DRO objectives based on posterior-informed Bayesian ambiguity sets. By employing exponential family models, the resulting problems benefit from a dual formulation which allows for the optimisation of worst-case risk criteria. This property, combined with generality across the entire exponential family, makes the approach particularly attractive for decision-making problems with statistical models. Since our dual is a single-stage stochastic program rather than a two-stage program, our framework offers computational advantages, making it feasible for larger-scale applications.

Understanding the growth of the ambiguity set volume as a function of the tolerance level $\epsilon$ would aid the interpretation of the robustness properties of these sets and is left for future work. Moreover, although both DRO-BAS formulations showcased improved empirical performance and computational advantages compared to existing methodologies, a theoretical analysis to identify conditions, beyond what we discussed in this work, favouring one formulation over the other would be highly beneficial.

Moreover, DRO-BAS$_{PP}$ allows for the use of models beyond the exponential family, as long as a closed-form posterior predictive is available. On the other hand, the dual formulation of DRO-BAS$_{PE}$ takes advantage of the properties of the exponential family models since the underlying nominal distribution is also a member. This is in contrast to BAS$_{PP}$ which can lead to general forms of posterior predictive distributions, outside the exponential family, leading to assumption violations in the derivation of the dual problem (Section 3.1). This advantage of BAS$_{PE}$ further offers a more efficient optimisation objective for the commonly used (see e.g. the Portfolio problem of Section 4.2) linear cost functions with a Normal likelihood (see Section 3.5).

Future work will aim to generalise DRO-BAS$_{PE}$ to models beyond the exponential family as it is possible that this will lead to differently shaped ambiguity sets, beyond KL-based ambiguity sets with a fixed nominal distribution. These might exhibit benefits in higher dimensions where the size of KL-based ambiguity sets can grow very fast with the tolerance level, leading to over-conservative decisions. Moreover, it would be valuable to theoretically examine the effect of the Monte Carlo sampling size on the out-of-sample cost, similar to our empirical analysis in Section 4. Such an analysis has already been performed for the Wasserstein distance and the $\chi^2$-divergence by Iyengar et al. (2023) but has not yet been extended to the KL-divergence.

Extending the BAS framework to general $\phi$-divergences, which have been previously used in DRO (e.g. Bayraksan & Love, 2015; Duchi & Namkoong, 2019), represents another promising direction for future work. In particular, analysing DRO-BAS$_{PP}$ would require deriving the dual formulation using convex conjugate results for $\phi$-divergences, though obtaining results on the tolerance level and worst-case distribution may prove more difficult due to the properties of the posterior predictive and the divergence function. The DRO-BAS$_{PE}$ setting is even more challenging, as it involves characterising the expected $\phi$-divergence under the posterior in closed form, which is critical for deriving its convex conjugate and establishing strong duality. Such derivations, as seen with the KL divergence, are key to obtaining results on tolerance level selection, tractable objectives in specific settings, and the form of the worst-case distribution. We believe addressing these challenges could lead to new, principled BAS formulations based on a broader class of divergences.

Finally, while DRO-BAS achieves worst-case robustness, it remains inherently tied to the Bayesian posterior. This dependence makes it vulnerable to severe model misspecification. Future work will investigate alternative posterior formulations that are more robust to model misspecification, to enhance the BAS construction and produce more robust posterior-informed ambiguity sets.

## Acknowledgments

CD acknowledges support from EPSRC grant [EP/T51794X/1] as part of the Warwick CDT in Mathematics and Statistics, EPSRC grant [EP/Y022300/1] and the G-research travel grant. PO and TD acknowledge support from a UKRI Turing AI acceleration Fellowship [EP/V02678X/1] and a Turing Impact Award from the Alan Turing Institute. For the purpose of open access, the authors have applied a Creative Commons Attribution (CC-BY) license to any Author Accepted Manuscript version arising from this submission.

## Impact Statement

This paper presents work whose goal is to advance the field of Machine Learning. There are many potential societal consequences of our work, none which we feel must be specifically highlighted here.

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

# Supplementary Material

The Supplementary Material is organised as follows: Appendix A provides details for all the conjugate exponential family examples used in the experiments, while Appendix B contains the proofs of all mathematical results appearing in the main text. Additional computational details are discussed in Appendix C and further details on the relationship between DRO-BAS$_{PE}$ and DRO-BAS$_{PP}$ are provided in Appendix D. Finally, Appendix E provides additional experimental details and results.

## A. Special Cases of Exponential Family Models

We derive the values of $G(\widehat{\tau}, \widehat{\nu})$ and $\hat{\eta}$ appearing in Theorem 3.6 for different likelihoods and conjugate prior/posterior pairs used in the experiments.

### A.1. Gaussian Model with Unknown Mean and Variance

This section considers a Bayesian model that estimates the unknown mean and variance of a univariate Gaussian distribution. We consider the natural parametrisation of the Normal distribution with unknown mean and unknown variance, written in its exponential family form and derive the conjugate prior and posterior parameters as well as $\hat{\eta}$ and $G(\widehat{\tau}, \widehat{\nu})$ in Theorem 3.6.

Consider the Gaussian likelihood for mean $\mu$ and precision $\lambda$. Then the likelihood of dataset $\mathcal{D}$ is:

$$p(\mathcal{D} \mid \mu, \lambda) = (2\pi)^{-\frac{n}{2}} \lambda^{\frac{n}{2}} \exp\left(-\frac{\sqrt{\lambda}}{2} \sum_{i=1}^{n} (\xi_i - \mu)^2\right). \tag{23}$$

The natural parameters and log-partition function of the normal distribution are

$$\eta = \begin{bmatrix} \eta_1 \\ \eta_2 \end{bmatrix} = \begin{bmatrix} \mu\lambda \\ -\frac{\lambda}{2} \end{bmatrix}, \quad A(\eta) = -\frac{\eta_1^2}{4\eta_2} - \frac{1}{2}\ln(-2\eta_2)$$

respectively. Before proceeding, we revisit the gamma and digamma functions.

**Definition A.1.** The gamma function $\Gamma : \mathbb{N} \to \mathbb{R}$ and digamma function $\psi : \mathbb{N} \to \mathbb{R}$ are

$$\Gamma(z) \triangleq (z-1)! \qquad\qquad\qquad \psi(z) \triangleq \frac{\mathrm{d}}{\mathrm{d}z}\ln\Gamma(z).$$

We are now ready to prove the following Corollary, which is a special case of Theorem 3.6.

**Corollary A.2.** *When the likelihood is a Gaussian distribution with unknown mean $\mu$ and precision $\lambda$ as in (23), and the conjugate prior and posterior are normal-gamma distributions with parameters $(\breve{\mu}, \breve{\kappa}, \breve{\alpha}, \breve{\beta})$ and $(\widehat{\mu}, \widehat{\kappa}, \widehat{\alpha}, \widehat{\beta})$ respectively, then Theorem 3.6 holds with $\hat{\eta} = \left(\frac{\widehat{\mu}(\widehat{\kappa}+1)}{2\widehat{\beta}}, -\frac{\widehat{\kappa}(\widehat{\kappa}+1)}{4\widehat{\beta}\widehat{\kappa}}\right)$ and $G(\widehat{\tau}, \widehat{\nu}) = \frac{1}{2}\left(\ln\widehat{\alpha} - \psi(\widehat{\alpha}) + \frac{1}{\widehat{\kappa}}\right)$.*

*Proof.* The prior distribution which leads to a conjugate posterior is the Normal-Gamma distribution with parameters hyper-parameters $\breve{\mu}, \breve{\kappa}, \breve{\alpha}, \breve{\beta}$:

$$\pi(\mu, \lambda \mid \breve{\mu}, \breve{\kappa}, \breve{\alpha}, \breve{\beta}) = NG(\mu, \lambda \mid \breve{\mu}, \breve{\kappa}, \breve{\alpha}, \breve{\beta})$$

$$= N(\mu \mid \breve{\mu}, (\breve{\kappa}\lambda)^{-1})Ga(\lambda \mid \breve{\alpha}, \breve{\beta})$$

$$= \frac{1}{Z_{NG}} \lambda^{\breve{\alpha}-\frac{1}{2}} \exp\left(-\breve{\beta}\lambda + \frac{\breve{\kappa}\lambda}{2}(\mu - \breve{\mu})^2\right)$$

where the normalisation constant $Z_{NG}$ is given by:

$$Z_{NG} = \breve{\beta}^{-\breve{\alpha}} \breve{\kappa}^{-1/2} \Gamma(\breve{\alpha})\sqrt{2\pi}.$$

Following the form of the conjugate prior in Definition 3.4 we can write the prior for $\eta$ as:

$$\pi(\eta \mid \breve{\tau}, \breve{\nu}) = \frac{1}{Z(\breve{\tau}, \breve{\nu})} \exp\left(\begin{bmatrix} \breve{\kappa}\breve{\mu} \\ \breve{\kappa}\breve{\mu}^2 + 2\breve{\beta} \end{bmatrix}^\top \begin{bmatrix} \eta_1 \\ \eta_2 \end{bmatrix} - \breve{\kappa}\left(-\frac{1}{2}\ln(-2\eta_2) - \frac{1}{4}\eta_1^2\eta_2^{-1}\right)\right)$$

$$= \frac{1}{Z(\breve{\tau}, \breve{\nu})} \exp\left(\breve{\tau}^\top \eta - \breve{\nu} A(\eta)\right).$$

where we set $\breve{\alpha} \triangleq \frac{\breve{\kappa}+1}{2}$ and it follows that:

$$\breve{\tau} = \begin{bmatrix} \breve{\tau}_1 \\ \breve{\tau}_2 \end{bmatrix} = \begin{bmatrix} \breve{\kappa}\breve{\mu} \\ \breve{\kappa}\breve{\mu}^2 + 2\breve{\beta} \end{bmatrix}, \quad \breve{\nu} = \breve{\kappa}, \quad Z(\breve{\tau}, \breve{\nu}) = \left(\frac{\breve{\tau}_2}{2} - \frac{\breve{\tau}_1^2}{2\breve{\nu}}\right)^{-\frac{\breve{\nu}+1}{2}} \breve{\nu}^{-\frac{1}{2}} \Gamma(\frac{\breve{\nu}+1}{2})\sqrt{2\pi}. \tag{24}$$

By conjugacy it follows that the posterior distribution for $\eta$ with parameters $\widehat{\tau} = \breve{\tau} + s(\mathcal{D})$ and $\widehat{\nu} = \breve{\nu} + n$, for sufficient statistic $s(\mathcal{D}) = \left(\sum_{i=1}^n \xi_i, \ \sum_{i=1}^n \xi^2\right)^\top$, is:

$$\Pi(\eta \mid \mathcal{D}, \widehat{\tau}, \widehat{\nu}) = \frac{1}{Z(\widehat{\tau}, \widehat{\nu})} \exp\left(\widehat{\tau}^\top \eta - \widehat{\nu} A(\eta)\right).$$

We can now derive $G(\widehat{\tau}, \widehat{\nu})$ for the univariate Normal with Normal-Gamma prior using Lemma 3.5. First,

$$\frac{\partial}{\partial \widehat{\nu}}\left(-\ln Z(\widehat{\tau}, \widehat{\nu})\right) = \frac{\partial}{\partial \widehat{\nu}}\left(\frac{\widehat{\nu}+1}{2}\ln\left(\frac{\widehat{\tau}_2}{2} - \frac{\widehat{\tau}_1^2}{2\breve{\nu}}\right) + \frac{1}{2}\ln\widehat{\nu} - \ln\Gamma\left(\frac{\breve{\nu}+1}{2}\right)\right)$$

$$= \frac{1}{2}\ln\left(\frac{\widehat{\tau}_2}{2} - \frac{\widehat{\tau}_1^2}{2\breve{\nu}}\right) + \frac{\widehat{\nu}+1}{2}\frac{\frac{\widehat{\tau}_1^2}{2\widehat{\nu}^2}}{\frac{\widehat{\tau}_2}{2} - \frac{\widehat{\tau}_1^2}{2\widehat{\nu}}} + \frac{1}{2\widehat{\nu}} - \frac{1}{2}\psi\left(\frac{\widehat{\nu}+1}{2}\right)$$

$$= \frac{1}{2}\ln\left(\frac{\widehat{\tau}_2}{2} - \frac{\widehat{\tau}_1^2}{2\breve{\nu}}\right) + \frac{(\widehat{\nu}+1)\widehat{\tau}_1^2}{2\widehat{\nu}(\widehat{\tau}_2\widehat{\nu} - \widehat{\tau}_1^2)} + \frac{1}{2\widehat{\nu}} - \frac{1}{2}\psi\left(\frac{\widehat{\nu}+1}{2}\right)$$

$$= \frac{1}{2}\ln\widehat{\beta} + \frac{(\widehat{\kappa}+1)\widehat{\kappa}^2\widehat{\mu}^2}{2\widehat{\kappa}(2\widehat{\beta}\widehat{\kappa})} + \frac{1}{2\kappa} - \frac{1}{2}\psi\left(\frac{\widehat{\kappa}+1}{2}\right)$$

$$= \frac{1}{2}\ln\widehat{\beta} + \frac{(\widehat{\kappa}+1)\widehat{\mu}^2}{4\widehat{\beta}} + \frac{1}{2\kappa} - \frac{1}{2}\psi\left(\frac{\widehat{\kappa}+1}{2}\right).$$

Furthermore, we have:

$$\hat{\eta} \triangleq -\nabla_{\widehat{\tau}}(-\ln Z(\widehat{\tau}, \widehat{\nu})) = -\nabla_{\widehat{\tau}}\left(\frac{\widehat{\nu}+1}{2}\ln\left(\frac{\widehat{\tau}_2}{2} - \frac{\widehat{\tau}_1^2}{2\breve{\nu}}\right) + \frac{1}{2}\ln\widehat{\nu} - \ln\Gamma\left(\frac{\breve{\nu}+1}{2}\right)\right)$$

$$= \left(\frac{\widehat{\tau}_1(\widehat{\nu}+1)}{\widehat{\tau}_2\widehat{\nu} - \widehat{\tau}_1^2}, -\frac{\widehat{\nu}(\widehat{\nu}+1)}{2(\widehat{\tau}_2\widehat{\nu} - \widehat{\tau}_1^2)}\right)$$

$$= \left(\frac{\widehat{\mu}(\widehat{\kappa}+1)}{2\widehat{\beta}}, -\frac{\widehat{\kappa}+1}{4\widehat{\beta}}\right).$$

Plugging this into the definition of the log-partition function we obtain:

$$A(\hat{\eta}) = -\frac{\hat{\eta}_1^2}{4\hat{\eta}_2} - \frac{1}{2}\ln(-2\hat{\eta}_2) = \frac{\widehat{\mu}^2(\widehat{\kappa}+1)}{4\widehat{\beta}} - \frac{1}{2}\ln\left(\frac{\widehat{\kappa}+1}{2\widehat{\beta}}\right)$$

and from Lemma 3.5 it follows that:

$$G(\widehat{\tau}, \widehat{\nu}) = \frac{\partial}{\partial \widehat{\nu}}\left(-\ln Z(\widehat{\tau}, \widehat{\nu})\right) - A(\hat{\eta})$$

$$= \frac{1}{2}\ln\widehat{\beta} + \frac{(\widehat{\kappa}+1)\widehat{\mu}^2}{4\widehat{\beta}} + \frac{1}{2\kappa} - \frac{1}{2}\psi\left(\frac{\widehat{\kappa}+1}{2}\right) - \frac{\widehat{\mu}^2(\widehat{\kappa}+1)}{4\widehat{\beta}} + \frac{1}{2}\ln\left(\frac{\widehat{\kappa+1}}{2\widehat{\beta}}\right)$$

$$= \frac{1}{2\widehat{\kappa}} - \frac{1}{2}\psi(\widehat{\alpha}) + \frac{1}{2}\ln\widehat{\alpha}.$$

Furthermore, for our implementation we can reparametrise $\hat{\eta}$ back to standard parameterization to obtain:

$$\hat{\lambda} = -2\hat{\eta}_2 = \frac{\widehat{\alpha}}{\widehat{\beta}}, \quad \hat{\mu} = \frac{\hat{\eta}_1}{\hat{\lambda}} = \widehat{\mu}.$$

$\square$

**Tolerance level** $\epsilon$  In the well-specified case, where we assume that $\mathbb{P}^\star \triangleq \mathbb{P}_\theta^\star$ for some $\theta^\star \in \Theta$, it is easy to obtain the required size of the ambiguity set exactly. Let $\theta^\star \triangleq (\mu^\star, \lambda^{\star-1})$ and $\mathbb{P}^\star \triangleq N(\mu^\star, \lambda^{\star-1})$. Using Corollary A.2 we obtain:

$$\epsilon_{\mathrm{PE}}^\star(n) = \mathbb{E}_{\mu,\lambda \sim NG(\mu,\lambda|\widehat{\mu},\widehat{\kappa},\widehat{\alpha},\widehat{\beta})}\left[d_{\mathrm{KL}}(\mathbb{P}^\star, \mathcal{N}(\xi \mid \mu, \lambda^{-1}))\right]$$

$$= d_{\mathrm{KL}}\left(\mathbb{P}^\star \parallel \mathcal{N}\left(\widehat{\mu}, \frac{\widehat{\beta}}{\widehat{\alpha}}\right)\right) + \frac{1}{2}\left(\frac{1}{\widehat{\kappa}} + \ln\widehat{\alpha} - \psi(\widehat{\alpha})\right)$$

$$= \ln\left(\sqrt{\lambda^\star\frac{\widehat{\beta}}{\widehat{\alpha}}}\right) + \frac{\lambda^{\star-1} + (\mu^\star - \widehat{\mu})^2}{2\frac{\widehat{\beta}}{\widehat{\alpha}}} - \frac{1}{2} + \frac{1}{2}\left(\frac{1}{\widehat{\kappa}} + \ln\widehat{\alpha} - \psi(\widehat{\alpha})\right)$$

$$= \frac{1}{2}\left(\ln\left(\lambda^\star\widehat{\beta}\right) + \frac{\lambda^{\star-1} + (\mu^\star - \widehat{\mu})^2}{\frac{\widehat{\beta}}{\widehat{\alpha}}} - 1 + \frac{1}{\widehat{\kappa}} - \psi(\widehat{\alpha})\right).$$

**Posterior predictive distribution**  For completeness, we remind the reader of the posterior predictive distribution, which is needed for the implementation of DRO-BAS$_{\mathrm{PP}}$. Under the setting of Corollary A.2, the posterior predictive distribution is the following Student-t distribution [3]:

$$\mathbb{P}_n \equiv t_{2\widehat{\alpha}}\left(\xi \mid \widehat{\mu}, \frac{\widehat{\beta}(\widehat{\kappa}+1)}{\widehat{\alpha}\widehat{\kappa}}\right).$$

## A.2. Exponential Likelihood with Conjugate Gamma Prior

We now consider the natural parametrisation of the Exponential distribution, written in its exponential family form and re-derive the conjugate prior and posterior parameters as well as $\hat{\eta}$ and $G(\widehat{\tau}, \widehat{\nu})$ in Theorem 3.6.

Consider the Gaussian likelihood of dataset $\mathcal{D}$ for rate parameter $\lambda > 0$:

$$p(\mathcal{D} \mid \lambda) = \exp\left(-\lambda\sum_{i=1}^n \xi_i + n\ln\lambda\right).$$

The natural parameter and log-partition function of the Exponential distribution are:

$$\eta = -\lambda, \quad A(\eta) = -\ln(-\eta)$$

respectively. We prove the following Corollary.

---

[3]For a full derivation see for example https://www.cs.ubc.ca/~murphyk/Papers/bayesGauss.pdf

**Corollary A.3.** *When the likelihood is an exponential distribution with rate parameter $\lambda > 0$ and a gamma prior and posterior pair is used with parameters $(\breve{\alpha}, \breve{\beta})$ and $(\widehat{\alpha}, \widehat{\beta})$ respectively, then Theorem 3.6 holds with $\hat{\eta} = -\frac{\widehat{\alpha}}{\widehat{\beta}}$ and* $G(\widehat{\tau}, \widehat{\nu}) = \ln \widehat{\alpha} - \psi(\widehat{\alpha})$.

*Proof.* The prior distribution which leads to a conjugate posterior is the Gamma distribution with hyper-parameters $\breve{\alpha}, \breve{\beta}$:

$$\pi(\lambda \mid \breve{\alpha}, \breve{\beta}) = \frac{1}{Z_G} \exp\left( -\breve{\beta}\lambda + (\breve{\alpha} - 1)\ln \lambda \right)$$

where the normalisation constant is $Z_G = \breve{\beta}^{-\breve{\alpha}} \Gamma(\breve{\alpha})$. By following the form of the conjugate prior in Definition 3.4 we can write the prior of $\eta$ as:

$$\pi(\eta \mid \breve{\tau}, \breve{\nu}) = \frac{1}{Z(\breve{\tau}, \breve{\nu})} \exp\left( \breve{\tau}\eta - \breve{\nu} A(\eta) \right)$$

where $\breve{\tau} = \breve{\beta}$, $\breve{\nu} = \breve{\alpha} - 1$ and $Z(\breve{\tau}, \breve{\nu}) = \breve{\tau}^{\breve{\nu}+1}\Gamma(\breve{\nu} + 1)$. By conjugacy, it follows that the posterior distribution for $\eta$ with parameters $\widehat{\tau} = \breve{\tau} + s(\mathcal{D})$ and $\widehat{\nu} = \breve{\nu} + n$, for sufficient statistic $s(\mathcal{D}) = \sum_{i=1}^{n} \xi_i$, is:

$$\Pi(\eta \mid \mathcal{D}, \widehat{\tau}, \widehat{\nu}) = \frac{1}{Z(\widehat{\tau}, \widehat{\nu})} \exp\left( \widehat{\tau}\eta - \widehat{\nu} A(\eta) \right).$$

We are now ready to derive $\hat{\eta}$ and $G(\widehat{\tau}, \widehat{\nu})$ for this example. Firstly,

$$\frac{\partial}{\partial \widehat{\nu}}(-\ln Z(\widehat{\tau}, \widehat{\nu})) = \frac{\partial}{\partial \widehat{\nu}}((\widehat{\nu} + 1)\ln \widehat{\tau} - \ln \Gamma(\widehat{\nu} + 1))$$

$$= \ln \widehat{\tau} - \psi(\widehat{\nu} + 1)$$

$$= \ln \widehat{\beta} - \psi(\widehat{\alpha}).$$

Furthermore, we have

$$\hat{\eta} \triangleq -\frac{\partial}{\partial \widehat{\tau}}(-\ln Z(\widehat{\tau}, \widehat{\nu})) = -\frac{\widehat{\nu} + 1}{\widehat{\tau}} = -\frac{\widehat{\alpha}}{\widehat{\beta}}$$

and

$$A(\hat{\eta}) = -\ln(-\hat{\eta}) = -\ln\left( \frac{\widehat{\alpha}}{\widehat{\beta}} \right).$$

Using Lemma 3.5 we obtain:

$$G(\widehat{\tau}, \widehat{\nu}) = \frac{\partial}{\partial \widehat{\nu}}(-\ln Z(\widehat{\tau}, \widehat{\nu})) - A(\hat{\eta}) = \ln \widehat{\beta} - \psi(\widehat{\alpha}) + \ln\left( \frac{\widehat{\alpha}}{\widehat{\beta}} \right) = \ln \widehat{\alpha} - \psi(\widehat{\alpha}).$$

Furthermore, for our implementation, we can reparametrise $\hat{\eta}$ to the standard parameterization and obtain:

$$\hat{\lambda} = -\hat{\eta} = \frac{\widehat{\alpha}}{\widehat{\beta}}.$$

$\square$

**Tolerance level** $\epsilon$   In the well-specified case, where we assume that $\mathbb{P}^\star \triangleq \mathbb{P}_\theta^\star$ for some $\theta^\star \in \Theta$, it is easy to obtain the required size of the ambiguity set exactly. Let $\theta^\star$ be the true rate parameter, i.e. $\mathbb{P}^\star \triangleq \mathrm{Exp}(\theta^\star)$. Using Corollary A.3 we obtain:

$$
\begin{aligned}
\epsilon_{\mathrm{PE}}^\star(n) &= \mathbb{E}_{\mathrm{Ga}(\theta | \widehat{\alpha}, \widehat{\beta})} \left[ d_{\mathrm{KL}}(\mathbb{P}^\star, \mathrm{Exp}(\theta) \right] \\
&= d_{\mathrm{KL}} \left( \mathbb{P}^\star \,\|\, \mathrm{Exp}\left( \frac{\widehat{\alpha}}{\widehat{\beta}} \right) \right) + \psi(\widehat{\alpha}) - \ln(\widehat{\alpha}) \\
&= \ln(\theta^\star) - \ln\left( \frac{\widehat{\alpha}}{\widehat{\beta}} \right) + \frac{\widehat{\alpha}}{\widehat{\beta}\theta^\star} - 1 + \psi(\widehat{\alpha}) - \ln(\widehat{\alpha}).
\end{aligned}
$$

**Posterior predictive distribution**   For completeness, we remind the reader of the posterior predictive distribution, which is needed for the implementation of DRO-BAS$_{\mathrm{PP}}$. Under the setting of Corollary A.3, the posterior predictive distribution is the following Lomax (Pareto type II) distribution:

$$
\mathbb{P}_n \equiv \mathrm{Lomax}(\widehat{\alpha}, \widehat{\beta}).
$$

### A.3. Multivariate Normal Likelihood with Normal-Wishart Prior

Now we assume the random variable $\xi$ is multivariate such that $\Xi \subseteq \mathbb{R}^D$ where $D$ is the dimension of the random variable. The definitions of the likelihood, prior, and posterior in terms of the natural parameters can be found in Chapter 3.4.4.3 of Murphy (2023). The likelihood is a multivariate normal distribution $\mathcal{N}(\xi \mid \mu, \Sigma)$ with $\mu \in \mathbb{R}^D$ and $\Sigma \in \mathbb{S}_+^D$ (where set $\mathbb{S}_+^D$ is the space of $D \times D$ positive semi-definite matrices):

$$
p(\xi \mid \mu, \Sigma) = (2\pi)^{-\frac{D}{2}} |\Sigma|^{-\frac{1}{2}} \exp\left( -\frac{1}{2}(\xi - \mu)^\top \Sigma^{-1}(\xi - \mu) \right). \tag{25}
$$

The natural parameters of the multivariate normal distribution are

$$
\eta = \begin{bmatrix} \eta_1 \\ \eta_2 \end{bmatrix} = \begin{bmatrix} \Sigma^{-1}\mu \\ -\frac{1}{2}\,\mathrm{vec}\left( \Sigma^{-1} \right) \end{bmatrix} \tag{26}
$$

where $\mathrm{vec}$ is a function that converts a matrix into a vector[4]. For convenience, we abuse notation and treat $\eta_2$ and $\tau_2$ as matrices instead of using their vectorised form.

We prove the following Corollary.

**Corollary A.4.** *When the likelihood is a multivariate Gaussian distribution with unknown mean $\mu \in \mathbb{R}^D$ and covariance matrix $\Sigma \in \mathbb{S}_+^D$ as in (25), and the conjugate prior and posterior are normal-Inverse-Wishart distributions with parameters $(\breve{\mu}, \breve{\kappa}, \breve{\iota}, \breve{\Psi})$ and $(\widehat{\mu}, \widehat{\kappa}, \widehat{\iota}, \widehat{\Psi})$ respectively, then Theorem 3.6 holds with*

$$
\hat{\eta} = \begin{bmatrix} (\widehat{\kappa} - D - 2)\widehat{\Psi}^{-1}\widehat{\mu} \\ -\frac{1}{2}(\widehat{\kappa} - D - 2)\widehat{\Psi}^{-\top} \end{bmatrix}
$$

*and*

$$
G(\widehat{\tau}, \widehat{\nu}) = -\frac{D}{2}\ln(\widehat{\kappa} - D - 2) - \frac{1}{2}\ln\left| \widehat{\Psi}^{-1} \right| + \frac{1}{2}(\widehat{\kappa} - D - 2)\widehat{\mu}^\top \widehat{\Psi}^{-1}\widehat{\mu}.
$$

*Proof.* The conjugate prior to the likelihood is the normal-inverse-Wishart (NIW) denoted by $\mathrm{NIW}\left( \mu, \Sigma \mid \breve{\mu}, \breve{\kappa}, \breve{\iota}, \breve{\Psi} \right)$.

The hyperparameters of the NIW have the following interpretation (Murphy, 2023): $\breve{\mu} \in \mathbb{R}^D$ is the prior mean for $\mu$, and

---

[4]Strictly speaking, since the upper diagonal of the covariance matrix is the same as the lower diagonal, our vectorised exponential family representation is not minimal. It is, however, more convenient and easier to work with.

$\breve{\kappa} \in \mathbb{R}_+$ quantifies how strongly we believe in this prior; matrix $\breve{\Psi} \in \mathbb{S}_{++}^D$ is (proportional to) the prior mean of $\Sigma$, and $\breve{\iota} \in \mathbb{R}_+$ dictates how strongly we believe in this prior. We can re-write the prior $\pi\left(\mu, \Sigma \mid \breve{\mu}, \breve{\kappa}, \breve{\iota}, \breve{\Psi}\right)$ in the form of an exponential family $\pi(\eta \mid \breve{\tau}, \breve{\nu})$: starting from the definition of the normal-inverse-Wishart distribution, we have

$$
\begin{aligned}
\mathrm{NIW}\left(\mu, \Sigma \mid \breve{\mu}, \breve{\kappa}, \breve{\iota}, \breve{\Psi}\right) &\triangleq \mathcal{N}\left(\mu \mid \breve{\mu}, \frac{1}{\breve{\kappa}}\Sigma\right) \times \mathrm{IW}\left(\Sigma \mid \breve{\Psi}, \breve{\iota}\right) \\
&= \frac{1}{Z_{\mathrm{NIW}}}|\Sigma|^{-\frac{1}{2}} \exp\left(-\frac{\breve{\kappa}}{2}(\mu - \breve{\mu})^\top \Sigma^{-1}(\mu - \breve{\mu})\right) \times |\Sigma|^{-\frac{1}{2}(\breve{\iota}+D+1)} \exp\left(-\frac{1}{2}\mathrm{Tr}\left(\breve{\Psi}\Sigma^{-1}\right)\right) \\
&= \frac{1}{Z_{\mathrm{NIW}}} \exp\left(-\frac{\breve{\kappa}}{2}\mu\Sigma^{-1}\mu + \breve{\kappa}\breve{\mu}^\top\Sigma^{-1}\mu - \frac{\breve{\kappa}}{2}\breve{\mu}^\top\Sigma\breve{\mu} - \frac{1}{2}(\iota + D + 2)\ln|\Sigma| - \frac{1}{2}\mathrm{Tr}\left(\breve{\Psi}\Sigma^{-1}\right)\right) \\
&= \frac{1}{Z_{\mathrm{NIW}}} \exp\left(\frac{\breve{\kappa}}{4}\eta_1^\top\eta_2^{-1}\eta_1 + \breve{\kappa}\eta_1^\top\breve{\mu} + \breve{\kappa}\breve{\mu}^\top\eta_2\breve{\mu} + \frac{1}{2}(\iota + D + 2)\ln|-2\eta_2| + \mathrm{Tr}\left(\breve{\Psi}\eta_2\right)\right) \\
&= \frac{1}{Z_{\mathrm{NIW}}} \exp\left(\begin{bmatrix}\breve{\kappa}\breve{\mu} \\ \breve{\kappa}\breve{\mu}\breve{\mu}^\top + \breve{\Psi}\end{bmatrix}^\top \begin{bmatrix}\eta_1 \\ \eta_2\end{bmatrix} - \breve{\nu}\left(-\frac{1}{2}\ln|-2\eta_2| - \frac{1}{4}\eta_1^\top\eta_2^{-1}\eta_1\right)\right) \\
&= \frac{1}{Z(\breve{\tau}, \breve{\nu})} \exp\left(\breve{\tau}^\top\eta - \breve{\nu}A(\eta)\right) = \pi(\eta \mid \breve{\tau}, \breve{\nu}).
\end{aligned}
$$

In the above, we recognise that $A(\eta) = -\frac{1}{2}\ln|-2\eta_2| - \frac{1}{4}\eta_1^\top\eta_2^{-1}\eta_1$ is the log partition function of the multivariate normal likelihood; we set $\breve{\nu} = \breve{\kappa} = \breve{\iota} + D + 2$; and we define $\breve{\tau}$ as

$$
\breve{\tau} = \begin{bmatrix}\breve{\tau}_1 \\ \breve{\tau}_2\end{bmatrix} = \begin{bmatrix}\breve{\kappa}\breve{\mu} \\ \mathrm{vec}\left(\breve{\kappa}\breve{\mu}\breve{\mu}^\top + \breve{\Psi}\right)\end{bmatrix} \tag{27}
$$

(similarly to $\eta_2$, we abuse notation and treat $\tau_2$ as a matrix). The normalisation constant $Z_{\mathrm{NIW}}$ for the NIW distribution may be written in terms of $\breve{\tau}$ and $\breve{\nu}$ as:

$$
\begin{aligned}
Z_{\mathrm{NIW}} &\triangleq 2^{\breve{\iota}D/2}\Gamma_D\left(\breve{\iota}/2\right)(2\pi/\breve{\kappa})^{D/2}|\breve{\Psi}|^{-\breve{\iota}/2} \\
&= 2^{D(\breve{\nu}-D-2)/2}\Gamma_D\left(\frac{\breve{\nu}-D-2}{2}\right)(2\pi/\breve{\nu})^{D/2}\left|\breve{\tau}_2 - \frac{1}{\breve{\nu}}\breve{\tau}_1\breve{\tau}_1^\top\right|^{-(\breve{\nu}-D-2)/2} \triangleq Z(\breve{\tau}, \breve{\nu})
\end{aligned} \tag{28}
$$

where $\Gamma_D$ is the multivariate gamma function with dimension $D$.

The posterior distribution is

$$
\Pi(\eta \mid \mathcal{D}) = \Pi(\eta \mid \widehat{\tau}, \widehat{\nu}) = \frac{1}{Z(\widehat{\tau}, \widehat{\nu})} \exp\left(\widehat{\tau}^\top\eta - \widehat{\nu}A(\eta)\right).
$$

The posterior update is $\widehat{\tau} = \breve{\tau} + s(\mathcal{D})$ and $\widehat{\nu} = \breve{\nu} + n$ where $s(\mathcal{D}) = \left(\sum_{i=1}^n \xi_i, \ \sum_{i=1}^n \xi\xi^\top\right)^\top$. Since the prior and posterior have the same form, the normalisation constant $Z(\widehat{\tau}, \widehat{\nu})$ is defined in the same way as (28).

Our goal is to derive the function $G(\widehat{\tau}, \widehat{\nu})$ from Lemma 3.5. First, we derive $\frac{\partial}{\partial \widehat{\nu}}\left(-\ln Z(\widehat{\tau}, \widehat{\nu})\right)$. We have

$$
\begin{aligned}
\frac{\partial}{\partial \widehat{\nu}}\left(-\ln Z(\widehat{\tau}, \widehat{\nu})\right) &= \frac{\partial}{\partial \widehat{\nu}}\Bigg(-\frac{D}{2}(\widehat{\nu}-D-2)\ln 2 - \ln \Gamma_D\left(\frac{(\widehat{\nu}-D-2)}{2}\right) - \frac{D}{2}\ln 2\pi \\
&\qquad + \frac{D}{2}\ln\widehat{\nu} + \frac{1}{2}(\widehat{\nu}-D-2)\ln\left|\widehat{\tau}_2 - \frac{1}{\widehat{\nu}}\widehat{\tau}_1\widehat{\tau}_1^\top\right|\Bigg) \\[2mm]
&= -\frac{D}{2}\ln 2 - \frac{1}{2}\psi_D\left(\frac{\widehat{\nu}-D-2}{2}\right) + \frac{D}{2\widehat{\nu}} + \frac{1}{2}\ln\left|\widehat{\tau}_2 - \frac{1}{\widehat{\nu}}\widehat{\tau}_1\widehat{\tau}_1^\top\right| \\
&\qquad + \frac{1}{2}(\widehat{\nu}-D-2)\operatorname{Tr}\left(\left(\widehat{\tau}_2 - \frac{1}{\widehat{\nu}}\widehat{\tau}_1\widehat{\tau}_1^\top\right)^{-1}\left(\widehat{\nu}^{-2}\widehat{\tau}_1\widehat{\tau}_1^\top\right)\right) \\[2mm]
&= -\frac{D}{2}\ln 2 - \frac{1}{2}\psi_D\left(\frac{\widehat{\kappa}-D-2}{2}\right) + \frac{D}{2\widehat{\kappa}} + \frac{1}{2}\ln\left|\widehat{\kappa}\widehat{\mu}\widehat{\mu}^\top + \widehat{\Psi} - \frac{\widehat{\kappa}^2}{\widehat{\kappa}}\widehat{\mu}\widehat{\mu}^\top\right| \\
&\qquad + \frac{\widehat{\kappa}-D-2}{2}\operatorname{Tr}\left(\left(\widehat{\kappa}\widehat{\mu}\widehat{\mu}^\top + \widehat{\Psi} - \frac{\widehat{\kappa}^2}{\widehat{\kappa}}\widehat{\mu}\widehat{\mu}^\top\right)^{-1}\left(\frac{\widehat{\kappa}^2}{\widehat{\kappa}^2}\widehat{\mu}\widehat{\mu}^\top\right)\right) \\[2mm]
&= -\frac{D}{2}\ln 2 - \frac{1}{2}\psi_D\left(\frac{\widehat{\kappa}-D-2}{2}\right) + \frac{D}{2\widehat{\kappa}} + \frac{1}{2}\ln\left|\widehat{\Psi}\right| + \frac{\widehat{\kappa}-D-2}{2}\widehat{\mu}^\top\widehat{\Psi}^{-1}\widehat{\mu}.
\end{aligned}
$$

The first equality above holds by definition of $Z(\widehat{\tau}, \widehat{\nu})$ and by log identities. The second equality holds by chain rule for differentiation and the identity $\partial(\ln|A|) = \operatorname{Tr}(A^{-1}\partial A)$ from (43) of Petersen & Pedersen (2008). In the third equality, we have substituted $\widehat{\tau}, \widehat{\nu}$ for $\widehat{\mu}, \widehat{\kappa}, \widehat{\Psi}$. In the last equality, we have simplified and used the fact that $\operatorname{Tr}(\widehat{\Psi}^{-1}\widehat{\mu}\widehat{\mu}^\top) = \widehat{\mu}^\top\widehat{\Psi}^{-1}\widehat{\mu}$.
Next, we need to evaluate $A(\hat{\eta})$. We note that by Lemma 3.5, $\hat{\eta}$ is

$$
\begin{aligned}
\hat{\eta} \triangleq \mathbb{E}_{\Pi(\eta|\widehat{\tau}, \widehat{\nu})}[\eta] &= -\nabla_{\widehat{\tau}}(-\ln Z(\widehat{\tau}, \widehat{\nu})) \\
&= -\nabla_{\widehat{\tau}}\Bigg(-\frac{D}{2}(\widehat{\nu}-D-2)\ln 2 - \ln \Gamma_D\left(\frac{(\widehat{\nu}-D-2)}{2}\right) - \frac{D}{2}\ln 2\pi \\
&\qquad + \frac{D}{2}\ln\widehat{\nu} + \frac{1}{2}(\widehat{\nu}-D-2)\ln\left|\widehat{\tau}_2 - \frac{1}{\widehat{\nu}}\widehat{\tau}_1\widehat{\tau}_1^\top\right|\Bigg) \\
&= -\nabla_{\widehat{\tau}}\left(\frac{1}{2}(\widehat{\nu}-D-2)\ln\left|\widehat{\tau}_2 - \frac{1}{\widehat{\nu}}\widehat{\tau}_1\widehat{\tau}_1^\top\right|\right).
\end{aligned}
$$

Before we obtain the partial derivative with respect to $\widehat{\tau}_1$ we need the following lemma.

**Lemma A.5.** *Let $x \in \mathcal{X}$ and $A \in \mathbb{R}^{n\times n}$. Then $\frac{\partial}{\partial x}\ln|A - xx^\top| = -2(A - xx^\top)^{-1}x$.*

*Proof.*

$$
\begin{aligned}
\frac{\partial}{\partial x} \ln |A - xx^\top| &\stackrel{(i)}{=} \mathrm{Tr}\left(-(A - xx^\top)^{-1} \frac{\partial(xx^\top)}{\partial x}\right) \\
&\stackrel{(ii)}{=} \mathrm{Tr}\left(-(A - xx^\top)^{-1}(x(\partial x)^\top + (\partial x)x^\top)\right) \\
&\stackrel{(iii)}{=} \mathrm{Tr}\left(-(A - xx^\top)^{-1}x(\partial x)^\top\right) + \mathrm{Tr}\left(-(A - xx^\top)^{-1}(\partial x)x^\top\right) \\
&\stackrel{(iv)}{=} \mathrm{Tr}\left(-(A - xx^\top)^{-1}x(\partial x)^\top\right) + \mathrm{Tr}\left(-(A - xx^\top)^{-1}x(\partial x)^\top\right) \\
&= -2\,\mathrm{Tr}\left((A - xx^\top)^{-1}x(\partial x)^\top\right) \\
&\stackrel{(v)}{=} -2(A - xx^\top)^{-1}x.
\end{aligned}
$$

where (i) follows from the Jacobi's formula (see e.g. Petersen & Pedersen, 2008, Equation 43) which says that $\partial \ln |X| = \mathrm{Tr}(X^{-1}\partial X)$, (ii) follows from the chain rule, (iii) follows from the property of trace that says $\mathrm{Tr}(A+B) = \mathrm{Tr}(A) + \mathrm{Tr}(B)$, (iv) follows from the fact that $x(\partial x)^\top$ is symmetric and finally $(v)$ follows from the property $\mathrm{Tr}(Auv^\top) = v^\top Au$. $\qquad\square$

We are now ready to derive the partial derivative of $-\ln Z(\widehat{\tau}, \widehat{\nu})$ with respect to $\widehat{\tau}_1$ using the above Lemma:

$$
\frac{\partial}{\partial \widehat{\tau}_1}\left(\frac{1}{2}(\widehat{\nu} - D - 2)\ln\left|\widehat{\tau}_2 - \frac{1}{\widehat{\nu}}\widehat{\tau}_1\widehat{\tau}_1^\top\right|\right) = -\frac{1}{\widehat{\nu}}(\widehat{\nu} - D - 2)(\widehat{\tau}_2 - \frac{1}{\widehat{\nu}}\widehat{\tau}_1\widehat{\tau}_1^\top)^{-1}\widehat{\tau}_1.
$$

where we used the fact that $\partial \ln |X| = \mathrm{Tr}(X^{-1}\partial X)$. Using the fact that for a square, non-singular matrix $X$: $\frac{\partial}{\partial X}\ln|X| = X^{-\top}$ (Petersen & Pedersen, 2008, Equation 49), the partial derivative with respect to $\widehat{\tau}_2$ is:

$$
\frac{\partial}{\partial \widehat{\tau}_2}\left(\frac{1}{2}(\widehat{\nu} - D - 2)\ln\left|\widehat{\tau}_2 - \frac{1}{\widehat{\nu}}\widehat{\tau}_1\widehat{\tau}_1^\top\right|\right) = \frac{1}{2}(\widehat{\nu} - D - 2)\left(\widehat{\tau}_2 - \frac{1}{\widehat{\nu}}\widehat{\tau}_1\widehat{\tau}_1^\top\right)^{-\top}.
$$

Hence,

$$
\hat{\eta} = \begin{bmatrix} \frac{1}{\widehat{\nu}}(\widehat{\nu} - D - 2)(\widehat{\tau}_2 + \frac{1}{\widehat{\nu}}\widehat{\tau}_1\widehat{\tau}_1^\top)^{-1}\widehat{\tau}_1 \\ -\frac{1}{2}(\widehat{\nu} - D - 2)\left(\widehat{\tau}_2 + \frac{1}{\widehat{\nu}}\widehat{\tau}_1\widehat{\tau}_1^\top\right)^{-\top} \end{bmatrix} = \begin{bmatrix} (\widehat{\kappa} - D - 2)\widehat{\Psi}^{-1}\widehat{\mu} \\ -\frac{1}{2}(\widehat{\kappa} - D - 2)\widehat{\Psi}^{-\top} \end{bmatrix}.
$$

From the definition of $A(\eta)$ for the multivariate normal distribution, we have

$$
\begin{aligned}
A(\hat{\eta}) &\triangleq -\frac{1}{2}\ln|-2\hat{\eta}_2| - \frac{1}{4}\hat{\eta}_1^\top \hat{\eta}_2^{-1}\hat{\eta}_1 \\
&= -\frac{1}{2}\ln\left|(\widehat{\kappa} - D - 2)\widehat{\Psi}^{-\top}\right| - \frac{1}{4}(-2)(\widehat{\kappa} - D - 2)\widehat{\mu}^\top\widehat{\Psi}^{-\top}\Psi^\top\widehat{\Psi}^{-1}\widehat{\mu} \\
&= -\frac{1}{2}\ln\left(\left|(\widehat{\kappa} - D - 2)I\right|\left|\widehat{\Psi}^{-\top}\right|\right) + \frac{1}{2}(\widehat{\kappa} - D - 2)\widehat{\mu}^\top\widehat{\Psi}^{-1}\widehat{\mu} \\
&= -\frac{D}{2}\ln(\widehat{\kappa} - D - 2) - \frac{1}{2}\ln\left|\widehat{\Psi}^{-1}\right| + \frac{1}{2}(\widehat{\kappa} - D - 2)\widehat{\mu}^\top\widehat{\Psi}^{-1}\widehat{\mu}.
\end{aligned}
$$

We have all the ingredients for calculating $G(\widehat{\tau}, \widehat{\nu})$:

$$
\begin{aligned}
G(\widehat{\tau}, \widehat{\nu}) &\triangleq \frac{\partial}{\partial \widehat{\nu}}\left(-\ln Z(\widehat{\tau}, \widehat{\nu})\right) - A(\hat{\eta}) \\
&= -\frac{D}{2}\ln 2 - \frac{1}{2}\psi_D\left(\frac{\widehat{\kappa} - D - 2}{2}\right) + \frac{D}{2\widehat{\kappa}} + \frac{1}{2}\ln\left|\widehat{\Psi}\right| + \frac{\widehat{\kappa} - D - 2}{2}\widehat{\mu}^\top \widehat{\Psi}^{-1}\widehat{\mu} \\
&\quad + \frac{D}{2}\ln(\widehat{\kappa} - D - 2) + \frac{1}{2}\ln\left|\widehat{\Psi}^{-1}\right| - \frac{1}{2}(\widehat{\kappa} - D - 2)\widehat{\mu}^\top\widehat{\Psi}^{-1}\widehat{\mu} \\
&= -\frac{D}{2}\ln 2 - \frac{1}{2}\psi_D\left(\frac{\widehat{\kappa} - D - 2}{2}\right) + \frac{D}{2\widehat{\kappa}} + \frac{D}{2}\ln(\widehat{\kappa} - D - 2)
\end{aligned}
$$

where we used the fact that $|\widehat{\Psi}^{-1}| = |\widehat{\Psi}|^{-1}$ Finally, for our implementation, we need to get the parameters $\hat{\mu}$, $\hat{\Sigma}$ of the normal likelihood from the parameter $\hat{\eta}$. Using equation (26), we have

$$
\begin{aligned}
\hat{\Sigma} &= (-2\hat{\eta}_2)^{-1} = \frac{1}{\widehat{\kappa} - D - 2}\widehat{\Psi} \\
\hat{\mu} &= \hat{\Sigma}\hat{\eta}_1 = \widehat{\mu}.
\end{aligned}
\tag{29}
$$

$\square$

**Tolerance level $\epsilon$**  In the well-specified case, where we assume that $\mathbb{P}^\star \triangleq \mathbb{P}_\theta^\star$ for some $\theta^\star \in \Theta$, it is easy to obtain the required size of the ambiguity set exactly. Let $\theta^\star \triangleq (\mu^\star, \Sigma^\star) \in \mathbb{R}^D \times \mathbb{S}_+^D$ and $\mathbb{P}^\star \triangleq N(\mu^\star, \Sigma^\star)$. Further let $\hat{\mu}$ and $\hat{\Sigma}$ as in (29). Using Corollary A.4 we obtain:

$$
\begin{aligned}
\epsilon_{\text{PE}}^\star(n) &= \mathbb{E}_{\mu, \Sigma \sim NIW(\mu, \Sigma|\widehat{\mu}, \widehat{\kappa}, \widehat{\iota}, \widehat{\Psi})}\left[d_{\text{KL}}(\mathbb{P}^\star, \mathcal{N}(\xi \mid \mu, \Sigma))\right] \\
&= d_{\text{KL}}\left(\mathbb{P}^\star \parallel \mathcal{N}\left(\hat{\mu}, \hat{\Sigma}\right)\right) - \frac{D}{2}\ln 2 - \frac{1}{2}\psi_D\left(\frac{\widehat{\kappa} - D - 2}{2}\right) + \frac{D}{2\widehat{\kappa}} + \frac{D}{2}\ln(\widehat{\kappa} - D - 2) \\
&= \frac{1}{2}\left[\log\frac{|\hat{\Sigma}|}{|\Sigma^\star|} - D + (\mu^\star - \hat{\mu})^\top\hat{\Sigma}^{-1}(\mu^\star - \hat{\mu}) + \text{Tr}\left\{\hat{\Sigma}^{-1}\Sigma^\star\right\}\right] \\
&\quad - \frac{D}{2}\ln 2 - \frac{1}{2}\psi_D\left(\frac{\widehat{\kappa} - D - 2}{2}\right) + \frac{D}{2\widehat{\kappa}} + \frac{D}{2}\ln(\widehat{\kappa} - D - 2).
\end{aligned}
$$

**Posterior predictive distribution**  For completeness, we remind the reader of the posterior predictive distribution, which is needed for the implementation of DRO-BAS$_{\text{PP}}$. Under the setting of Corollary A.4, the posterior predictive distribution is the following multivariate Student-t distribution (see e.g. Murphy, 2023, p. 96):

$$
\mathbb{P}_n \equiv \mathcal{T}_{\widehat{\kappa} - D + 1}\left(\xi \mid \widehat{\mu}, \frac{\widehat{\Psi}(\widehat{\kappa} + 1)}{\widehat{\kappa}(\widehat{\kappa} - D + 1)}\right).
$$

# B. Proofs of Theoretical Results

## B.1. Proof of Proposition 3.2

Before proving the required upper bound, we recall the definition of the KL divergence and its convex conjugate.

**Definition B.1** (KL-divergence). Let $\mu, \nu \in \mathcal{P}(\Xi)$ and assume $\mu$ is absolutely continuous with respect to $\nu$ ($\mu \ll \nu$). The KL divergence of $\mu$ with respect to $\nu$ is defined as:

$$
d_{\text{KL}}(\mu \| \nu) \triangleq \int_\Xi \ln\left(\frac{\mu(d\xi)}{\nu(d\xi)}\right)\mu(d\xi).
$$

**Lemma B.2** (Conjugate of the KL-divergence). *Let $\nu \in \mathcal{P}(\Xi)$ be non-negative and finite. The convex conjugate $d_{KL}^\star(\cdot\|\nu)$ of $d_{KL}(\cdot\|\nu)$ is*

$$d_{KL}^\star(\cdot\|\nu)(h) = \ln\left(\int_\Xi \exp(h)d\nu\right).$$

*Proof.* See Proposition 28 and Example 7 in Agrawal & Horel (2021). $\square$

We are now ready to prove Proposition 3.2.

*Proof.* We introduce a Lagrangian variable $\gamma \geq 0$ for the KL constraint on the left-hand side of (6) as follows:

$$\sup_{\mathbb{Q}:d_{KL}(Q\|\mathbb{P}_n)\leq\epsilon} \mathbb{E}_Q[f_x] \overset{(i)}{=} \inf_{\gamma\geq 0} \sup_{\mathbb{Q}\in\mathcal{P}(\Xi)} \mathbb{E}_Q[f_x] + \gamma\epsilon - \gamma d_{KL}(\mathbb{Q}\|\mathbb{P}_n)$$

$$\overset{(ii)}{=} \inf_{\gamma\geq 0} \gamma\epsilon + (\gamma d_{KL}(\cdot\|\mathbb{P}_n))^\star(f_x)$$

$$\overset{(iii)}{=} \inf_{\gamma\geq 0} \gamma\epsilon + \gamma\ln\mathbb{E}_{\mathbb{P}_n}\left[\exp\left(\frac{f_x}{\gamma}\right)\right].$$

Equality (i) holds by strong duality since $\mathbb{P}_n$ is a strictly feasible point to the primal constraint. Equality (ii) holds by the definition of the conjugate function. Finally, equality (iii) holds by Lemma B.2 and the fact that for $\gamma \geq 0$ and function $\phi$, $(\gamma\phi)^\star(y) = \gamma\phi^\star(y/\gamma)$. $\square$

## B.2. Proof of Proposition 3.3

*Proof.* The result follows from a standard Lagrangian duality argument and an application of Jensen's inequality. More specifically, we introduce a Lagrangian variable $\gamma \geq 0$ for the expected-ball constraint on the left-hand side of (13) as follows:

$$\sup_{\mathbb{Q}:\mathbb{E}_{\theta\sim\Pi}[d_{KL}(Q\|\mathbb{P}_\theta)]\leq\epsilon} \mathbb{E}_Q[f_x] \overset{(i)}{\leq} \inf_{\gamma\geq 0} \sup_{\mathbb{Q}\in\mathcal{P}(\Xi)} \mathbb{E}_Q[f_x] + \gamma\epsilon - \gamma\mathbb{E}_\Pi[d_{KL}(\mathbb{Q}\|\mathbb{P}_\theta)]$$

$$\overset{(ii)}{=} \inf_{\gamma\geq 0} \gamma\epsilon + \sup_{\mathbb{Q}\in\mathcal{P}(\Xi)} \mathbb{E}_Q[f_x] - \mathbb{E}_\Pi[\gamma d_{KL}(\mathbb{Q}\|\mathbb{P}_\theta)]$$

$$\overset{(iii)}{=} \inf_{\gamma\geq 0} \gamma\epsilon + (\mathbb{E}_\Pi[\gamma d_{KL}(\cdot\|\mathbb{P}_\theta)])^\star(f_x)$$

$$\overset{(iv)}{\leq} \inf_{\gamma\geq 0} \gamma\epsilon + \mathbb{E}_\Pi\left[(\gamma d_{KL}(\cdot\|\mathbb{P}_\theta))^\star(f_x)\right]$$

$$\overset{(v)}{=} \inf_{\gamma\geq 0} \gamma\epsilon + \mathbb{E}_\Pi\left[\gamma\ln\mathbb{E}_{\mathbb{P}_\theta}\left[\exp\left(\frac{f_x}{\gamma}\right)\right]\right].$$

Inequality (i) holds by weak duality. Equality (ii) holds by linearity of expectation and a simple rearrangement. Equality (iii) holds by the definition of the conjugate function. Inequality (iv) holds by Jensen's inequality $(\mathbb{E}[\cdot])^\star \leq \mathbb{E}[(\cdot)^\star]$ because the conjugate is a convex function. Equality (v) holds by Lemma B.2 and the fact that for $\gamma \geq 0$ and function $\phi$, $(\gamma\phi)^\star(y) = \gamma\phi^\star(y/\gamma)$. $\square$

## B.3. Proof of Lemma 3.5

We first give a closed-form expression for the expected log-likelihood under the posterior. We start by deriving an expression for the posterior mean, i.e. $\hat{\eta} \triangleq \mathbb{E}_{\Pi(\eta|\mathcal{D},\hat{\tau},\hat{\nu})}[\eta]$. We follow a similar technique as the one presented in Endres et al. (2022).

We start by using the fact that the posterior density integrates to 1 and take partial derivatives with respect to $\hat{\tau}$. Note that

by the differentiability assumptions and the fact that the posterior p.d.f. is Lebesgue-integrable, we can interchange the differentiability and integral operations:

$$\int \frac{1}{Z(\widehat{\tau},\widehat{\nu})} \exp\left(\widehat{\tau}^{\top}\eta - \widehat{\nu}A(\eta)\right) d\eta = 1$$

$$\Rightarrow \nabla_{\widehat{\tau}}\left(\int \frac{1}{Z(\widehat{\tau},\widehat{\nu})} \exp\left(\widehat{\tau}^{\top}\eta - \widehat{\nu}A(\eta)\right) d\eta\right) = 0$$

$$\Rightarrow \nabla_{\widehat{\tau}}\left(\frac{1}{Z(\widehat{\tau},\widehat{\nu})}\right) Z(\widehat{\tau},\widehat{\nu}) + \frac{1}{Z(\widehat{\tau},\widehat{\nu})}\int \eta \exp\left(\widehat{\tau}^{\top}\eta - \widehat{\nu}A(\eta)\right) d\eta = 0$$

$$\Rightarrow \mathbb{E}_{\Pi(\eta|\mathcal{D},\widehat{\tau},\widehat{\nu})}[\eta] = -\nabla_{\widehat{\tau}}\left(\frac{1}{Z(\widehat{\tau},\widehat{\nu})}\right) Z(\widehat{\tau},\widehat{\nu})$$

$$\Rightarrow \hat{\eta} = -\nabla_{\widehat{\tau}}\left(\frac{1}{Z(\widehat{\tau},\widehat{\nu})}\right) Z(\widehat{\tau},\widehat{\nu})$$

$$\Rightarrow \hat{\eta} = -\nabla\widehat{\tau}\left(-\ln Z(\widehat{\tau},\widehat{\nu})\right). \tag{30}$$

We now have an expression for $\hat{\eta}$ and we can hence write:

$$\begin{aligned}
\mathbb{E}_{\eta\sim p(\eta|\widehat{\tau},\widehat{\nu})}[\ln p(\xi\mid\eta)] &= \mathbb{E}_{\eta\sim p(\eta|\widehat{\tau},\widehat{\nu})}\left[\ln h(\xi) + \eta^{T}s(\xi) - A(\eta)\right] \\
&= \ln h(\xi) + \mathbb{E}_{\eta\sim p(\eta|\widehat{\tau},\widehat{\nu})}[\eta]^{T}s(\xi) - \mathbb{E}_{\eta\sim p(\eta|\widehat{\tau},\widehat{\nu})}[A(\eta)] \\
&= \ln h(\xi) + \hat{\eta}^{T}s(\xi) - \mathbb{E}_{\eta\sim p(\eta|\widehat{\tau},\widehat{\nu})}[A(\eta)] \\
&= \ln h(\xi) + (\hat{\eta})^{T}s(\xi) - A(\hat{\eta}) + A(\hat{\eta}) - \mathbb{E}_{\eta\sim p(\eta|\widehat{\tau},\widehat{\nu})}[A(\eta)] \\
&= \ln p(\xi\mid\hat{\eta}) + A(\hat{\eta}) - \mathbb{E}_{\eta\sim p(\eta|\widehat{\tau},\widehat{\nu})}[A(\eta)]
\end{aligned} \tag{31}$$

where in the last equality we used the exponential family form of the likelihood (see Definition 3.4). To obtain an expression for the expected log-partition function we follow a similar argument: we start from the posterior p.d.f. and differentiate with respect to $\widehat{\nu}$ under the differentiability and integrability assumptions:

$$\int_{\Omega} \frac{1}{Z(\widehat{\tau},\widehat{\nu})} \exp\left(\widehat{\tau}^{\top}\eta - \widehat{\nu}A(\eta)\right) d\eta = 1$$

$$\Rightarrow \frac{\partial}{\partial\widehat{\nu}}\left(\int_{\Omega} \frac{1}{Z(\widehat{\tau},\widehat{\nu})} \exp\left(\widehat{\tau}^{\top}\eta - \widehat{\nu}A(\eta)\right) d\eta\right) = 0$$

$$\Rightarrow \frac{\partial}{\partial\widehat{\nu}}\left(\frac{1}{Z(\widehat{\tau},\widehat{\nu})}\right)\int_{\Omega} \exp\left(\widehat{\tau}^{\top}\eta - \widehat{\nu}A(\eta)\right) d\eta + \frac{1}{Z(\widehat{\tau},\widehat{\nu})}\int_{\Omega} \frac{\partial}{\partial\widehat{\nu}} \exp\left(\widehat{\tau}^{\top}\eta - \widehat{\nu}A(\eta)\right) d\eta = 0$$

$$\Rightarrow \frac{\partial}{\partial\widehat{\nu}}\left(\frac{1}{Z(\widehat{\tau},\widehat{\nu})}\right) Z(\widehat{\tau},\widehat{\nu}) + \frac{1}{Z(\widehat{\tau},\widehat{\nu})}\int_{\Omega} -A(\eta)\exp\left(\widehat{\tau}^{\top}\eta - \widehat{\nu}A(\eta)\right) d\eta = 0$$

$$\Rightarrow \mathbb{E}_{\eta\sim\Pi(\eta|\mathcal{D},\widehat{\tau},\widehat{\nu})}[A(\eta)] = \frac{\partial}{\partial\widehat{\nu}}\left(\frac{1}{Z(\widehat{\tau},\widehat{\nu})}\right) Z(\widehat{\tau},\widehat{\nu})$$

$$\Rightarrow \mathbb{E}_{\eta\sim\Pi(\eta|\mathcal{D},\widehat{\tau},\widehat{\nu})}[A(\eta)] = \frac{\partial}{\partial\widehat{\nu}}\left(-\ln(Z(\widehat{\tau},\widehat{\nu}))\right). \tag{32}$$

By substituting Equation (32) in Equation (31) we obtain a closed-form expectation for the expected log-likelihood as

follows:

$$\mathbb{E}_{\eta \sim p(\eta | \widehat{\tau}, \widehat{\nu})}[\ln p(\xi \mid \eta)] = \ln p(\xi \mid \hat{\eta}) + A(\hat{\eta}) - \mathbb{E}_{\eta \sim p(\eta | \widehat{\tau}, \widehat{\nu})}[A(\eta)]$$

$$= \ln p(\xi \mid \hat{\eta}) - \left( \frac{\partial}{\partial \widehat{\nu}} \left( -\ln(Z(\widehat{\tau}, \widehat{\nu})) - A(\hat{\eta}) \right) \right)$$

$$\triangleq \ln p(\xi \mid \hat{\eta}) - G(\widehat{\tau}, \widehat{\nu}). \tag{33}$$

It is straightforward to show that $G(\widehat{\tau}, \widehat{\nu})$ is non-negative:

$$G(\widehat{\tau}, \widehat{\nu}) = \frac{\partial}{\partial \widehat{\nu}} \left( -\ln(Z(\widehat{\tau}, \widehat{\nu})) - A(\hat{\eta}) \right)$$

$$= \mathbb{E}_{\eta \sim \Pi(\eta | \mathcal{D}, \widehat{\tau}, \widehat{\nu})}[A(\eta)] - A\left( \mathbb{E}_{\eta \sim \Pi(\eta | \mathcal{D}, \widehat{\tau}, \widehat{\nu})}[\eta] \right)$$

$$\geq 0$$

where the equality follows from Equation (32) and the inequality follows from Jensen's inequality since the log-partition function of an exponential family likelihood is a convex function (Brown, 1986, Theorem 1.13). Finally, we can proceed with the main result. Starting from the left-hand side, we have

$$\mathbb{E}_{\eta \sim \Pi} [d_{\text{KL}}(\mathbb{Q} \| \mathbb{P}_\theta)] \overset{(i)}{=} \mathbb{E}_{\eta \sim \Pi(\eta | \mathcal{D}, \widehat{\tau}, \widehat{\nu})} \left[ \int_\Xi q(\xi) \ln \left( \frac{q(\xi)}{p(\xi \mid \eta)} \right) d\xi \right]$$

$$\overset{(ii)}{=} \mathbb{E}_{\eta \sim \Pi(\eta | \mathcal{D}, \widehat{\tau}, \widehat{\nu})} \left[ \int_\Xi q(\xi) \ln (q(\xi)) - q(\xi) \ln (p(\xi \mid \eta)) d\xi \right]$$

$$\overset{(iii)}{=} \int_\Xi q(\xi) \ln (q(\xi)) - q(\xi) \cdot \mathbb{E}_{\eta \sim \Pi(\eta | \mathcal{D}, \widehat{\tau}, \widehat{\nu})} [\ln (p(\xi \mid \eta))] d\xi$$

$$\overset{(iv)}{=} \int_\Xi q(\xi) \ln (q(\xi)) - q(\xi) \cdot \left( \ln p(\xi \mid \hat{\eta}) - G(\widehat{\tau}, \widehat{\nu}) \right) d\xi$$

$$\overset{(v)}{=} \int_\Xi q(\xi) \ln \left( \frac{q(\xi)}{p(\xi \mid \hat{\eta})} \right) d\xi + \int_\Xi q(\xi) \cdot G(\widehat{\tau}, \widehat{\nu}) d\xi$$

$$\overset{(vi)}{=} d_{\text{KL}}(\mathbb{Q} \| \mathbb{P}_{\hat{\eta}}) + G(\widehat{\tau}, \widehat{\nu}).$$

where (i) is by the definition of the KL-divergence; (ii) follows by $\log$ properties; (iii) holds by linearity of expectation; (iv) holds by (33); (v) holds by rearrangement and properties of $\log$ and (vi) holds by the definition of the KL-divergence.

### B.4. Proof of Theorem 3.6

We begin by restating the Lagrangian dual from the proof of Equation (13) for the exponential family case, but with the added claim that strong duality holds between the primal and dual problems:

$$\sup_{\mathbb{Q} : \mathbb{E}_{\eta \sim \Pi}[d_{\text{KL}}(\mathbb{Q} \| \mathbb{P}_\eta)] \leq \epsilon} \mathbb{E}_\mathbb{Q}[f_x] = \inf_{\gamma \geq 0} \sup_{\mathbb{Q} \in \mathcal{P}(\Xi)} \mathbb{E}_\mathbb{Q}[f_x] + \gamma \epsilon - \gamma \mathbb{E}_\Pi [d_{\text{KL}}(\mathbb{Q} \| \mathbb{P}_\eta)]. \tag{34}$$

*Proof.* The conditions under which our claim of strong duality holds will be proved later. Next, we substitute the right-hand side of equation (vi) above into the dual problem in (34):

$$\sup_{\mathbb{Q} : \mathbb{E}_{\eta \sim \Pi}[d_{\text{KL}}(\mathbb{Q} \| \mathbb{P}_\eta)] \leq \epsilon} \mathbb{E}_{\xi \sim \mathbb{Q}}[f_x(\xi)]$$

$$= \inf_{\gamma \geq 0} \gamma \epsilon + \sup_{\mathbb{Q} \in \mathcal{P}(\Xi)} \int_\Xi f_x(\xi) q(\xi) d\xi - \gamma \left( d_{\text{KL}} (\mathbb{Q} \| p(\xi \mid \hat{\eta})) + G(\widehat{\tau}, \widehat{\nu}) \right)$$

$$= \inf_{\gamma \geq 0} \gamma \epsilon - \gamma G(\widehat{\tau}, \widehat{\nu}) + (\gamma \, d_{\text{KL}} (\cdot \| p(\xi \mid \hat{\eta})))^\star (f_x(\xi))$$

$$= \inf_{\gamma \geq 0} \gamma(\epsilon - G(\widehat{\tau}, \widehat{\nu})) + \gamma \ln \mathbb{E}_{\xi \sim p(\xi | \hat{\eta})} \left[ \exp \left( \frac{f_x(\xi)}{\gamma} \right) \right],$$

where the second and third equality holds by the definition of the conjugate of the KL-divergence and by Lemma B.2.

Finally, it remains to argue that strong duality holds. First, note that the primal problem is a concave optimisation problem with respect to distribution $\mathbb{Q}$. Second, when $\epsilon > G(\widehat{\tau}, \widehat{\nu})$, then distribution $\hat{\mathbb{Q}} = p(\xi \mid \hat{\eta})$ is a strictly feasible point to the primal constraint because

$$\mathbb{E}_{\theta \sim \Pi}[d_{\mathrm{KL}}(\hat{\mathbb{Q}} \parallel \mathbb{P}_\theta)] = 0 < \epsilon - G(\widehat{\tau}, \widehat{\nu}).$$

$\square$

## C. Bayesian DRO Reformulation

We begin by recalling the *two-stage* stochastic optimisation proposed by Shapiro et al. (2023) for Bayesian DRO. Let $\mathcal{X}$ be the set of feasible decisions. The decision $x \in \mathcal{X}$ is the called the here-and-now decision: the decision is made before the realisation of the uncertainty of $\theta$. The first stage objective function is $H(x, \theta)$. The first-stage decision solves:

$$\min_{x \in \mathcal{X}} \mathbb{E}_{\theta \sim \Pi} [H(x, \theta)]. \tag{35}$$

The Lagrangian decision $\lambda$ is the second-stage decision: the decision is made after the realisation of the uncertainty of $\theta$. The second stage problem is:

$$H(x, \theta) := \inf_{\lambda > 0} \left\{ \lambda \epsilon + \lambda \ln \mathbb{E}_{\xi \sim p(\xi|\theta)} \left[ \exp \left( \frac{f_x(\xi)}{\lambda} \right) \right] \right\}. \tag{36}$$

We emphasise the two-stage nature of this program: the variable $\lambda$ is a function of $\theta$ and can only be optimised after the realisation of $\theta$.

Shapiro et al. (2023) solve the two-stage stochastic program with a nested SAA. The outer SAA samples $\theta_i \sim \pi(\theta | \hat{\xi}_1, \ldots, \hat{\xi}_N)$ where $i = 1, \ldots, N_\theta$. The inner SAA samples $\xi_j \sim p(\xi | \theta_i)$ where $j = 1, \ldots, N_\xi$. For a given $\theta_j$ and given decision $x \in \mathcal{X}$, the authors solve:

$$H(x, \theta) \approx \hat{G}(x, \theta) = \inf_{\lambda > 0} \left\{ \lambda \epsilon + \lambda \ln \frac{1}{N_\xi} \sum_{\xi_j \sim p(\xi|\theta_i)} \exp \left( \frac{f(x, \xi_j)}{\lambda} \right) \right\}. \tag{37}$$

The drawback of this method is the necessity to iterate over each decision $x$ in the set of feasible set $\mathcal{X}$. For example, in the one-dimensional Newsvendor Problem, the authors set $\mathcal{X} = \{x \in \mathbb{R} : 0 \leq x \leq 50\}$ and then iterate over $\mathcal{X}$ with a grid search:

1. For each $x = 0, 1, \ldots, 50$, evaluate (37). Let $x^\star$ be the minimum attained.

2. For each $x$ from $x^\star - 1$ to $x^\star + 1$ in increments of 0.01, evaluate (37) and return the minimum attained.

When the set $\mathcal{X}$ large, this strategy is clearly not scalable. Instead, we jointly minimise the decision variable and Lagrangian variables using an out-of-the-box commercial solver. Our algorithm relies upon the observation that the samples $\theta_i \sim \pi_N$ form a discrete, empirical distribution. Let $\lambda_i$ be the Lagrangian variable that depends upon sample $\theta_i$. From Section 2.3.1 of Shapiro et al. (2021), we can switch the expectation in (35) with the minimisation in (37) to obtain:

$$\inf_{x, \lambda_1, \ldots, \lambda_{N_\theta}} \left\{ \frac{1}{N_\theta} \sum_{i=1}^{N_\theta} \lambda_i \epsilon + \lambda_i \ln \frac{1}{N_\xi} \sum_{j=1}^{N_\xi} \exp \left( \frac{f(x, \xi_j^{(i)})}{\lambda_i} \right) : \lambda_1, \ldots, \lambda_{N_\theta} > 0, x \in \mathcal{X} \right\}, \tag{38}$$

where we have denoted the $j^{\mathrm{th}}$ sample from $p(\xi|\theta_i)$ by $\xi_j^{(i)}$. We now have a joint minimisation over the decision $x$ and Lagrangian variables $\lambda_i$ where $i = 1, \ldots, N_\theta$. However, the second term in (38) cannot yet be given directly to a solver. We make two transformations to (38) to get the minimisation problem into a standard form that can be recognised by a solver. The first transformation uses the log identity $\ln \frac{a}{b} = \ln a - \ln b$. The second transformation introduces $N_\theta \times N_\xi$

epigraphical variables $t_{ij}$ and constraints $f(x, \xi_j^{(i)}) \le t_{ij}$ for all $i = 1, \ldots, N_\theta$ and $j \in 1, \ldots, N_\xi$. Re-writing (38) with these two transformations, we have:

$$\inf_{x, \lambda_1, \ldots, \lambda_{N_\theta}} \quad \frac{1}{N_\theta} \sum_{i=1}^{N_\theta} \lambda_i \epsilon - \lambda \ln N_\xi + \lambda_i \ln \sum_{j=1}^{N_\xi} \exp\left(\frac{t_{ij}}{\lambda_i}\right)$$

$$\text{s. t.} \quad f(x, \xi_j^{(i)}) \le t_{ij}, \qquad i = 1, \ldots, N_\theta, j \in 1, \ldots, N_\xi$$

$$\lambda_1, \ldots, \lambda_{N_\theta} > 0, \quad x \in \mathcal{X} \tag{39}$$

To conclude, we observe that the third term is a perspective function of the form $\lambda_i \, \text{LSE}(t_i / \lambda_i)$, where $t_i = (t_{i1}, \ldots, t_{iN_\xi})$ and LSE is the Log-Sum-Exp function. As $\lambda_i \to 0$, we must take care in our implementation to avoid numerical instability. Fortunately, we can make use of Lemma C.1 in our optimisation algorithm, which says as $\lambda_i \to 0$, then

$$\lambda_i \, \text{LSE}(t_i / \lambda_i) \to \max(t_{i1}, \ldots, t_{iN_\xi}).$$

**Lemma C.1.** *Let* $\text{LSE}(y_1, \ldots, y_n)$ *be the Log-Sum-Exp function defined by*

$$\text{LSE}(y_1, \ldots, y_n) \triangleq \ln \sum_{i=1}^{n} \exp(y_i).$$

*The perspective of the Log-Sum-Exp function converges to the max function, that is:*

$$\lim_{\tau \to 0} \tau \, \text{LSE}\left(\frac{y_1}{\tau}, \ldots, \frac{y_n}{\tau}\right) = \max(y_1, \ldots, y_n).$$

*Proof.* Let $y^\star = \max(y_1, \ldots, y_n)$ and let $C$ the cardinality of the set of maximum elements:

$$C = |\{y_i : y_i = y^\star, i = 1, \ldots, n\}|.$$

The Log-Sum-Exp function has the following property:

$$\text{LSE}(y_1, \ldots, y_n) = y^\star + \text{LSE}(y_1 - y^\star, \ldots, y_n - y^\star).$$

Using this property, we have

$$\lim_{\tau \to 0} \tau \, \text{LSE}\left(\frac{y_1}{\tau}, \ldots, \frac{y_n}{\tau}\right) = y^\star + \lim_{\tau \to 0} \tau \ln \sum_{i=1}^{n} \exp\left(\frac{y_i - y^\star}{\tau}\right)$$

$$= y^\star + \lim_{\tau \to 0} \tau \ln C = y^\star.$$

To see why the second equality holds, notice that $y_i - y^\star \le 0$ for all $i = 1, \ldots, n$. Thus, we have

$$\lim_{\tau \to 0} \exp\left(\frac{y_i - y^\star}{\tau}\right) = \begin{cases} 0, & \text{if } y_i - y^\star < 0, \\ 1, & \text{if } y_i - y^\star = 0, \end{cases}$$

where we assume $0/0 = 0$ for the case of $y_i - y^\star = 0$. $\qquad \square$

**Linear objective and Normal likelihood.** Recall from Section 3.5 that when the objective function $f_x(\xi) = \xi^\top x$ and the likelihood is a Normal distribution $\mathcal{N}(\mu, \Sigma)$, then we can use the MGF to solve the optimisation problem over $\lambda$ in (37) to obtain

$$H(x, \theta) = H(x, \mu, \Sigma) = \mu^\top x + \sqrt{2\epsilon} \sqrt{x^\top \Sigma x}.$$

Assuming the posterior distribution is a Normal-inverse-Wishart (NIW) distribution, the BDRO dual from (35) can be re-written as

$$\min_{x \in \mathcal{X}} \mathbb{E}_{\mu, \Sigma \sim \text{NIW}}[H(x, \theta)] = \min_{x \in \mathcal{X}} \mathbb{E}_{\mu \sim \text{NIW}}[\mu^\top x] + \sqrt{2\epsilon} \, \mathbb{E}_{\Sigma \sim \text{NIW}}\left[\sqrt{x^\top \Sigma x}\right]. \tag{40}$$

The expectation over $\mu$ above can be computed in closed form, whilst the expectation over $\Sigma$ can be approximated by Monte Carlo sampling.

# D. Further Details on the Connection Between DRO-BAS Formulations

In this section, we provide the proof of Corollary 3.7 and make some additional comments on the relationship between the two formulations. We begin with Corollary 3.7.

*Proof of Corollary 3.7.* By (17), $\text{BAS}_{\text{PE}}$ is equivalent to a KL-based discrepancy set with nominal distribution $\mathbb{P}_{\hat{\eta}}$ and a posterior-adjusted constant $G(\hat{\tau}, \hat{\nu})$. To see this, consider some $\epsilon > 0$ and $\mathbb{Q} \in \mathcal{P}(\Xi)$ such that $\mathbb{Q} \ll \mathbb{P}_{\eta}$ for any $\eta \in \Omega$. Then by (17):

$$\mathbb{E}_{\eta \sim \Pi}[d_{\text{KL}}(\mathbb{Q}||\mathbb{P}_{\eta})] \leq \epsilon \Rightarrow d_{\text{KL}}(\mathbb{Q}, \mathbb{P}_{\hat{\eta}}) + G(\hat{\tau}, \hat{\nu}) \leq \epsilon$$
$$\Rightarrow d_{\text{KL}}(\mathbb{Q}, \mathbb{P}_{\hat{\eta}}) \leq \epsilon - G(\hat{\tau}, \hat{\nu}).$$

This means that the DRO-$\text{BAS}_{\text{PE}}$ problem in (12), restricted to the exponential family case, is equivalent to the following optimisation problem:

$$\min_{x} \sup_{\mathbb{Q} \in \mathcal{B}_{\epsilon - G(\hat{\tau}, \hat{\nu})}(\mathbb{P}_{\hat{\eta}})} \mathbb{E}_{\xi \sim \mathbb{Q}}[f_x(\xi)].$$

$\square$

This observation is specific to the exponential family and we should not expect this to hold for general model families and posteriors.

Another interesting connection between the two formulations regards cases where one constitutes the subset of the other. One such case is the following:

**Lemma D.1.** *For general model family $\mathcal{P}_{\Theta}$ and fixed $\epsilon \geq 0$, $BAS_{PE}$ is a subset of $BAS_{PP}$, i.e. $\mathcal{A}_{\epsilon}(\Pi) \subseteq \mathcal{B}_{\epsilon}(\mathbb{P}_n)$.*

Hence, for fixed $\epsilon$, if the DGP is contained in both ambiguity sets, then the DRO-$\text{BAS}_{\text{PE}}$ risk is upper bounded by the DRO-$\text{BAS}_{\text{PP}}$ risk resulting in DRO-$\text{BAS}_{\text{PP}}$ decisions being overly conservative.

*Proof of Lemma D.1.* This result follows from Jensen's inequality by noting that the KL is convex on the second argument and hence for any $\mathbb{Q} \in \mathcal{A}_{\epsilon}(\Pi)$ we have

$$d_{\text{KL}}(\mathbb{Q}||\mathbb{P}_n) = d_{\text{KL}}(\mathbb{Q}||\mathbb{E}_{\theta \sim \Pi}[\mathbb{P}_{\theta}]) \leq \mathbb{E}_{\theta \sim \Pi}[d_{\text{KL}}(\mathbb{Q}||\mathbb{P}_{\theta})] \leq \epsilon$$

and hence $\mathbb{Q} \in \mathcal{B}_{\epsilon}(\mathbb{P}_n)$. $\square$

In the exponential family case, this holds for all $\epsilon \geq \epsilon_{\text{PE}}^{\star}$, with $\epsilon_{\text{PE}}^{\star}$ as defined in (20). In particular, for all $\epsilon \geq \epsilon_{\text{PE}}^{\star}$ it follows that:

$$\mathbb{E}_{\xi \sim \mathbb{P}^{\star}}[f_x(\xi)] \leq \sup_{\mathbb{Q}: \mathbb{E}_{\eta \sim \Pi}[d_{\text{KL}}(\mathbb{Q}||\mathbb{P}_{\eta})] \leq \epsilon} \mathbb{E}_{\xi \sim \mathbb{Q}}[f_x(\xi)] \leq \sup_{\mathbb{Q}: d_{\text{KL}}(\mathbb{Q}||\mathbb{E}_{\eta \sim \Pi}[\mathbb{P}_{\eta}]) \leq \epsilon} \mathbb{E}_{\xi \sim \mathbb{Q}}[f_x(\xi)]$$

where the first inequality follows from (21) and the second inequality follows from Lemma D.1. However, it is worth noting that this is not guaranteed for values of $\epsilon$ smaller than $\epsilon_{\text{PE}}^{\star}$ so no definite conclusions can be made for the performance of the methods for different values of $\epsilon$ or for any $\epsilon \leq \epsilon_{\text{PE}}^{\star}$. Exploring this further would be an important aspect of future work. We further discuss computational differences of the two objectives in the next section and empirically investigate both methods in Section 4.

# E. Supplementary Experiments & Details

In this section, we provide additional details about the setup of experiments, the metrics used to evaluate the algorithms, and some supplementary experiments.

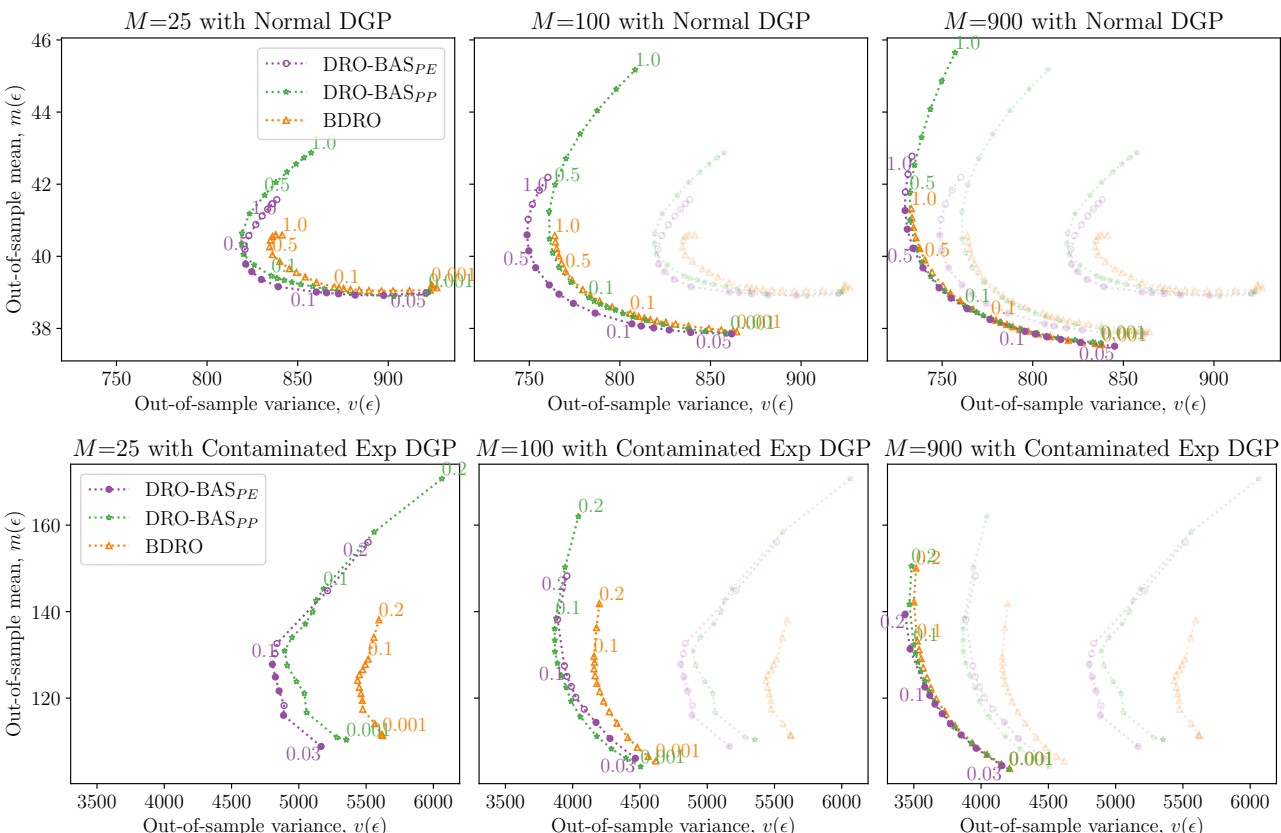

*Figure 4.* The out-of-sample mean-variance tradeoff for normal DGP (top row) and Contaminated Exponential DGP (bottom row). We vary $\epsilon$ for DRO-BAS$_{PE}$, DRO-BAS$_{PP}$, and BDRO when the total number of samples from the model is 25 (left), 100 (middle), and 900 (right) and plot the markers in bold. For illustration purposes, blurred markers/lines show the other smaller values of $M$; for example, the right column shows $M = 900$ in bold and $M = 25, 100$ in blurred. Each marker corresponds to a single value of $\epsilon$, some of which are labelled (e.g. $\epsilon = 0.1, 0.5, 3.0$). Filler markers indicate the given $\epsilon$ lies on the Pareto frontier of the model-based algorithms. If the marker is unfilled, this indicates the given $\epsilon$ is Pareto dominated by at least one other point.

**Implementation.** We implemented the dual problems for DRO-BAS (Theorem 3.6) and BDRO (Shapiro et al., 2023) in Python 3.11 using CVXPY version 1.5.2 (Diamond & Boyd, 2016; Agrawal et al., 2018) and the MOSEK solver version 10.1.28. Three machines were used to run experiments, each with a Dual Intel Xeon E5-2643 v3 @ 3.4 Ghz (12 cores/24 threads total) with 128GB RAM (5 GB per thread / 10GB per core). The SLURM workload manager was used to schedule jobs. Each job was allocated a single core and 10GB of RAM such that upto 12 jobs can be ran in parallel on a single machine.

**Out-of-sample mean and variance.** For a given $\epsilon$, we calculate the total out-of-sample mean $m(\epsilon)$ and variance $v(\epsilon)$ of the cost as:

$$m(\epsilon) = \frac{1}{JT} \sum_{j=1}^{J} \sum_{t=1}^{T} f(x_\epsilon^{(j)}, \xi_{n+t}),$$

$$v(\epsilon) = \frac{1}{JT-1} \sum_{j=1}^{J} \sum_{t=1}^{T} \left( f(x_\epsilon^{(j)}, \xi_{n+t}) - m(\epsilon) \right)^2,$$

where $x_\epsilon^{(j)}$ is the obtained solution on training dataset $\mathcal{D}_n^{(j)}$. The variance formula we use above is simply the total variance across all $T$ observations and all $J$ test datasets. For random seed $j = 1, \ldots, 500$, we sample $n = 20$ training observations $\mathcal{D}_n^{(j)}$ from the DGP, then sample $T = 50$ test points. This contrasts with the formula proposed in Gotoh et al. (2021) for a

| | | Solve time (seconds) | | | Total sample time (seconds $\times 10^{-4}$) | | |
|------|-----|---------------------|---------------------|---------------|---------------------|---------------------|---------------|
| DGP | $M$ | DRO-BAS$_{PE}$ | DRO-BAS$_{PP}$ | BDRO | DRO-BAS$_{PE}$ | DRO-BAS$_{PP}$ | BDRO |
| Exp | 25 | **0.024** (0.003) | **0.024** (0.003) | 0.068 (0.012) | **0.097** (0.007) | 0.117 (0.009) | 0.360 (0.018) |
| | 100 | **0.036** (0.003) | 0.037 (0.003) | 0.148 (0.020) | 0.099 (0.007) | 0.122 (0.015) | 0.614 (0.045) |
| | 900 | **0.422** (0.030) | 0.428 (0.032) | 0.724 (0.040) | 0.114 (0.010) | 0.155 (0.013) | 1.611 (0.117) |
| Tr. $\mathcal{N}$ | 25 | **0.024** (0.003) | 0.024 (0.003) | 0.067 (0.012) | 0.070 (0.006) | 0.138 (0.008) | 0.121 (0.009) |
| | 100 | **0.035** (0.003) | **0.035** (0.003) | 0.145 (0.020) | **0.072** (0.006) | 0.142 (0.009) | 0.156 (0.013) |
| | 900 | **0.398** (0.020) | 0.406 (0.021) | 0.683 (0.033) | 0.090 (0.008) | 0.185 (0.012) | 0.292 (0.025) |
| 5D $\mathcal{N}$ | 25 | 0.031 (0.004) | **0.030** (0.004) | 0.104 (0.012) | 0.423 (0.034) | 0.400 (0.023) | 4.279 (0.243) |
| | 100 | **0.080** (0.009) | **0.080** (0.010) | 0.210 (0.022) | 0.437 (0.033) | 0.422 (0.025) | 7.852 (0.342) |
| | 900 | **0.919** (0.097) | 0.948 (0.100) | 1.145 (0.072) | 0.525 (0.039) | 0.592 (0.040) | 22.150 (0.892) |

*Table 3.* Newsvendor Problem: Average solve (in seconds) and sample time (in seconds $\times 10^{-4}$) with the associated standard deviation in brackets. The Exponential, 1D Truncated Normal (T. $\mathcal{N}$) and 5D Normal DGPs correspond to the Newsvendor problem setting in Section 4.1. The time of the algorithm with the fastest solve is in bold; if two algorithms have the same solve time, then the sample time is in bold. The solve and sample times for the 1D well-specified Normal and contaminated Exponential DGPs in Figure 4 are very similar to Tr. $\mathcal{N}$ and 5D $\mathcal{N}$ respectively, and are thus omitted.

bootstrapping application which was also used by Shapiro et al. (2023) in their BDRO experiments.

### E.1. Newsvendor Problem - Additional Details

This subsection is dedicated to additional details and experiments that complement Section 4.1 on the Newsvendor Problem.

**Data-generating processes.** In the main text, we evaluated the methods on two well-specified settings: an exponential distribution with rate parameter $\lambda = \frac{1}{20}$ and a 5D multivariate normal distribution with mean $\mu_\star = (10, 20, 30, 35, 22)$ and full covariance $\Sigma_\star$. In this section, we also provide supplementary experiments for a univariate Normal distribution DGP with mean $\mu_\star = 25$ and standard deviation $\sigma_\star = 10$; the likelihood is also Normal, so the model is well-specified.

In Section 4.1, we also evaluated the algorithms when the likelihood is misspecified. The DGP is a Truncated Normal distribution with mean $\mu_\star = 10$, standard deviation $\sigma_\star = 10$, and truncation $\xi \in [0, \infty)$; the likelihood is a univariate Normal distribution. In this section, we provide another misspecified example. The DGP is a Contaminated Exponential distribution whilst the likelihood is an Exponential distribution with conjugate Gamma posterior/prior. 80% of the observations from the Contaminated Exponential DGP are sampled from an Exponential distribution with rate parameter $\lambda = \frac{1}{20}$; the other 20% of observations are sampled from a Normal distribution with mean 100 and standard deviation 0.5.

**Prior hyperparameters.** When the likelihood is a univariate Normal distribution (i.e. when the DGP is Normal or Truncated Normal), the prior and posterior are Normal-Gamma distributions; we set the prior hyperparameters to be $\breve{\mu} = 0$ and $\breve{\kappa}, \breve{\alpha}, \breve{\beta} = 1$. When the likelihood is an Exponential distribution, the prior and posterior are Gamma distributions with prior hyperparameters $\breve{\alpha}, \breve{\beta} = 1$. Finally, when the likelihood is a multivariate Normal distribution with dimension $D = 5$, the prior and posterior are Normal-Inverse-Wishart distributions with prior hyperparameters $\breve{\mu} = \mathbf{0} \in \mathbb{R}^D$, $\breve{\iota} = D + 1$, $\breve{\kappa} = \breve{\iota} + D + 2$, and $\breve{\Psi} = I_D$, where $I_D$ is the $D$-dimensional identity matrix.

**Supplementary experiment: Normal DGP.** The top row of Figure 4 shows that the conclusions of Section 4.1 hold for the well-specified case of a Normal DGP/likelihood, that is, for $M = 25, 100$, DRO-BAS Pareto dominates BDRO in the OOS mean-variance tradeoff, whilst performance is similar for $M = 900$.

**Supplementary experiment: Contaminated Exponential DGP.** The bottom row of Figure 4 shows misspecification context where the DGP is a Contaminated Exponential and the likelihood is an Exponential distribution. As $\epsilon$ increases, the OOS mean of DRO-BAS and BDRO increases fast but with little reduction in the OOS variance: a decision maker is unlikely to desire this property. This implies that we should use a very small value of $\epsilon$ because a larger $\epsilon$ does not give

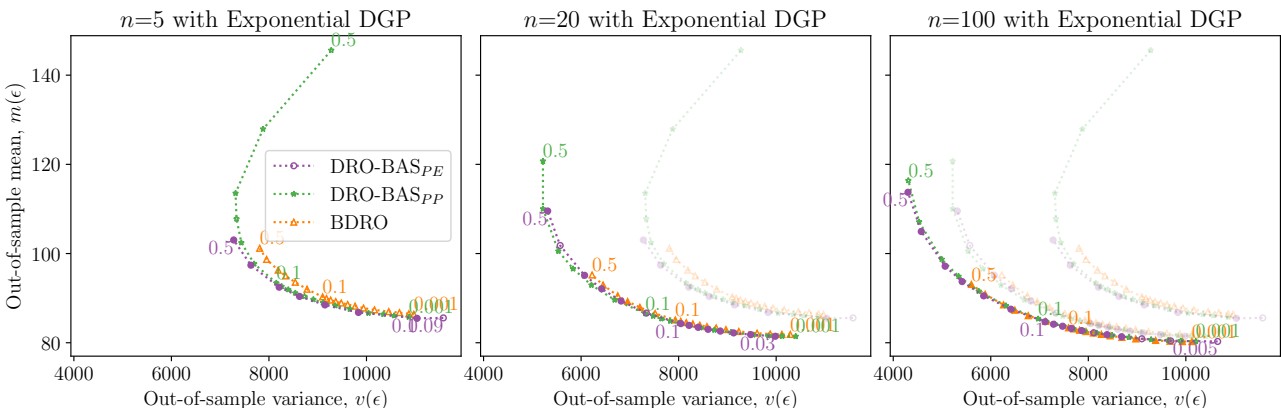

**Figure 5.** The out-of-sample mean-variance tradeoff for Exponential DGP when the total number of observations $n$ is 5 (left), 20 (middle), and 100 (right). For illustration purposes, blurred markers/lines show the previous two values of $n$; for example, the right column shows $n = 100$ in bold and $n = 5, 20$ in blurred. Each marker corresponds to a single value of $\epsilon$, some of which are labelled (e.g. $\epsilon = 0.1, 0.5, 3.0$). If the marker is filled, this indicates the given $\epsilon$ lies on the Pareto frontier. If the marker is unfilled, this indicates the given $\epsilon$ is Pareto dominated by at least one other point.

much more robustness. Contrast this to the other DGPs we have analysed where the mean-variance tradeoff creates a nice Pareto frontier curve on a subset of the $\epsilon$ values whereby an increase in the OOS mean is traded for a decrease in the OOS variance. In particular, contrast this to the well-specified Exponential example from the top row of Figure 2. All of this provides motivation for future work on Bayesian ambiguity that specifically target model misspecification.

**Solve and sample times.** Table 3 shows the solve times on the Newsvendor Problem for algorithms on different DGPs. As we noted in Section 4.1, the solve and sample times are broadly comparable for DRO-BAS and BDRO, although both DRO-BAS algorithms are faster across all DGPs and all sample sizes $M$.

**Results for varying number of observations.** We now examine the results from the Newsvendor problem for an increasing number of observations. We fix the number of MC samples to $M = 100$ and run the experiment for dataset size $n = 5, 20, 100$. Results for $n = 5, 20$ were also reported in Shapiro et al. (2023) for the well-specified Normal and Exponential DGPs/likelihoods. Figure 5 shows that as expected performance of both methods improves as the number of observations increases. Moreover, DRO-BAS continues to Pareto dominate or match BDRO in all cases with the difference between the two methods being bigger for a smaller number of observations.

**Why does the OOS variance increase as $\epsilon$ increases?** The behaviour of the OOS variance $v(\epsilon)$ in the Newsvendor experiments is due to the behaviour of the variance of the solution (denoted by $v_x(\epsilon)$). In turn, the variance of the solution is driven by the Newsvendor asymmetric cost function and its interplay with the worst-case distribution for each $\epsilon$. Figure 6 shows the behaviour of the solution for the Newsvendor problem under a Normal DGP corresponding to the top plot in Figure 4. The Newsvendor cost-function is piece-wise linear with 2 pieces, and hence, the true risk (expected cost under the DGP) has two different gradient slopes. The optimal solution lies at the intersection of the two pieces of the objective function, which coincides with the point with respect to $x$ where the true risk is zero. The piece corresponding to larger solutions has a significantly smaller slope (see 4th plot of Figure 6), leading to a smaller true risk.

Let's consider a fixed value of $M = 25$ in Figure 6. For small values of $\epsilon$, the variance of the solution $v_x(\epsilon)$ is smaller because fewer distributions are included in the ambiguity set; hence, the obtained solution is fairly stable over replications (see first plot of Figure 6). However, because BAS likely does not include the DGP, the decision is prone to risk; thus, the OOS variance of the *cost function* $v(\epsilon)$ is large (see the Pareto curve of the first plot of Figure 4). In this regime, the OOS variance (of the cost) reduces as we increase epsilon and better capture the DGP.

As $\epsilon$ increases further, the ambiguity sets begin to include more distributions, leading to the mean solution moving to values larger than the optimal solution where the slope of the true risk is smaller (see 4th plot of Figure 6). This makes sense because, as we increase $\epsilon$, we become more conservative. However, as $\epsilon$ increases to very large values (in this case > 0.5),

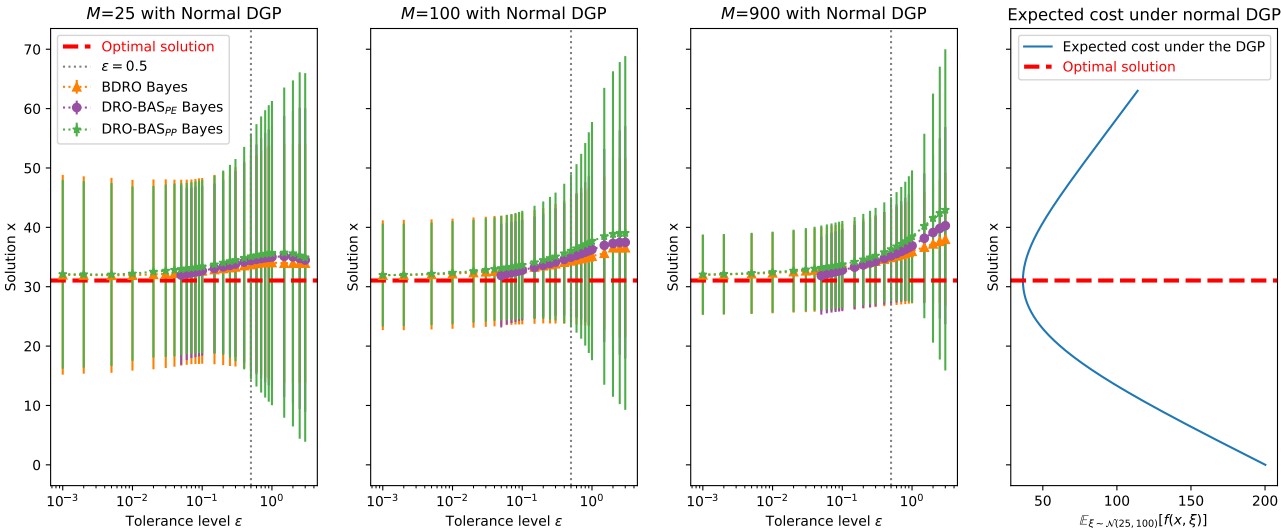

*Figure 6.* The first three plots showcase the mean solution (decision variable $x$) with the associated variance (error bars) obtained by the methods applied to the Newsvendor problem with a Normal DGP for $M = 25, 100, 900$. The last plot shows the true risk (expected cost under the DGP) for each value of the solution $x$. Red dotted lines show the optimal solution under the DGP.

the ambiguity set contains a lot of arbitrary distributions, significantly increasing the out-of-sample cost. The methods then push the solution towards smaller values of $x$ as can be observed by the big variance $v_x(\epsilon)$ on the first plot for larger values of $\epsilon$.

This behaviour is common for all values of $M$, however, there is an important distinction: for smaller values of $M$ (25, 100), the error bars extend way below and above the optimal solution. If we associate this with the form of the true risk on the last plot in Figure 6, we can expect a very high variance $v(\epsilon)$ of the out-of-sample cost. On the other hand, the variance of the solution $v_x(\epsilon)$ for large M ($M = 900$) does not extend below the optimal solution by a large degree. This means that as the optimisation becomes more exact, the methods suggest staying on values bigger than the optimal solution, which corresponds to the smaller slope of the true risk. This is why for $M = 900$ in the top row of Figure 4, the out-of-sample variance $v(\epsilon)$ does not increase by a lot for large values of $\epsilon$.

### E.2. Portfolio Problem: Additional Details

The dataset from Bruni et al. (2016) contains two baselines which we include for comparison in Figure 7: a Markowitz mean-variance risk model (Markowitz, 1952) and the DowJones index. In Figure 7, we show the cumulative returns on the test datasets. We note that all three DRO methods have a larger cumulative return than the Markowitz and DowJones index baselines.

**Sliding time window construction (Bruni et al., 2016).** The first train $\mathcal{D}_n^{(1)}$ and test $\mathcal{D}_T^{(1)}$ dataset contain the first 52 weeks of returns and following 12 weeks of returns respectively. We construct the second train dataset $\mathcal{D}_n^{(2)}$ by excluding the first 12 weeks of returns from $\mathcal{D}_n^{(1)}$ and including the 12 weeks from $\mathcal{D}_T^{(1)}$, whilst the second test dataset $\mathcal{D}_T^{(2)}$ contains the next 12 weeks of data following $\mathcal{D}_T^{(1)}$. We repeat this procedure for all $j = 1, \ldots, J$ until the end of the dataset.

### E.3. Discussion on Empirical vs Model-based DRO and Additional Comparisons

In this section, we further discuss the relationship between empirical and model-based DRO methods. This work focuses on scenarios where the decision-maker has access to both observations from the data-generating process (DGP) and a *model family* that describes the underlying data distribution. Specifically, we examine situations where incorporating uncertainty quantification about the model parameters into the worst-case minimization problem is crucial. This setting is particularly relevant when a model is required to capture a complex relationship between covariates and outcome variables, such as in regression tasks or Bayesian risk minimization, where the decision-maker seeks protection against a poor posterior

*Figure 7.* The Portfolio *cumulative* return across test datasets. For each DRO method, a line corresponds to a distinct $\epsilon$ value (shown in brackets in the legend). The vertical dotted line marks the start of the first test set on week 52.

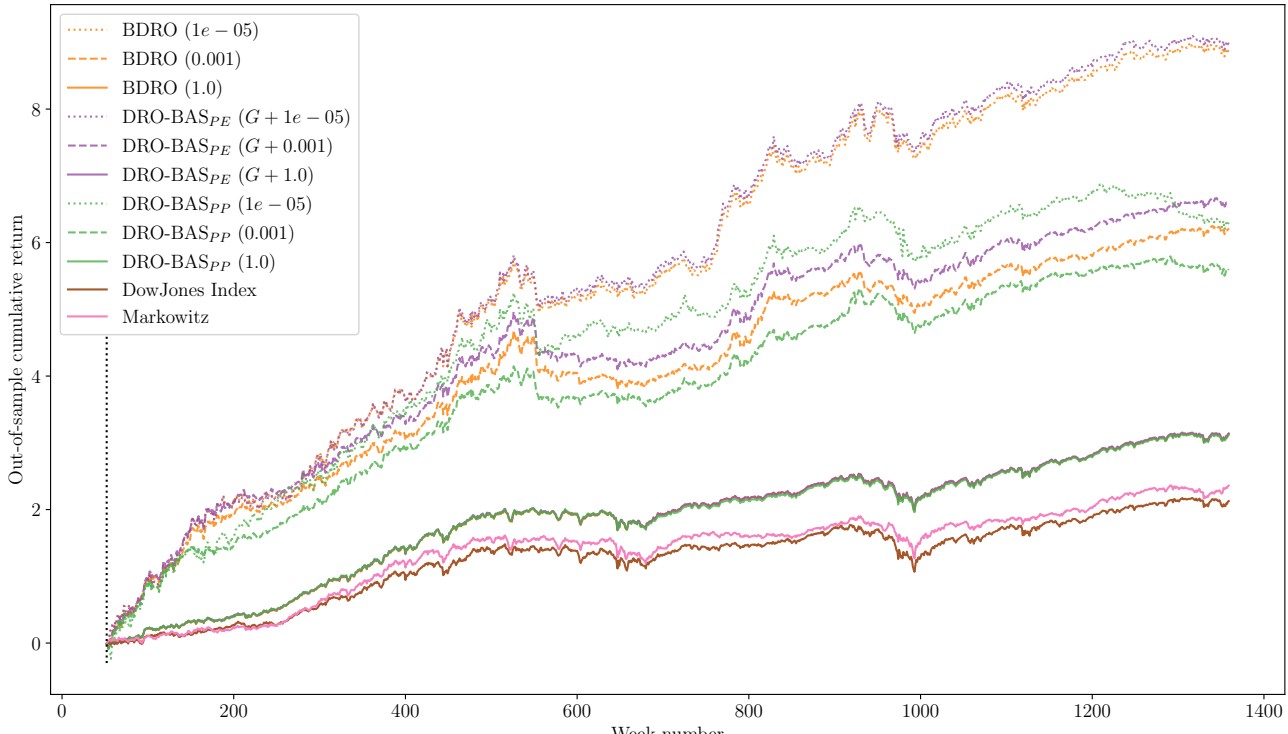

distribution arising from limited or noisy data. In such instances, purely data-driven (empirical) DRO formulations may not be the most suitable approach. To better understand the practical differences between these methods, we empirically assess our proposed approach against two empirical DRO methods. Firstly, the Empirical KL DRO framework introduced by Hu & Hong (2013) uses the following ambiguity set:

$$\mathcal{B}_\epsilon(\hat{\mathbb{P}}_n) = \{\mathbb{P} : d_{\text{KL}}(\mathbb{P}||\hat{\mathbb{P}}_n) \leq \epsilon\}.$$

Secondly, we compare against Wasserstein DRO with a $p = 2$ norm (see e.g. Kuhn et al., 2019; Gao & Kleywegt, 2023).

Figure 8 shows the OOS mean-variance trade-off of Empirical KL DRO (in gray with crosses) and Wasserstein DRO (in brown with plusses) versus the model-based DRO approaches of DRO-BAS and BDRO on the Newsvendor Problem for two DGPs. For a given DGP, note that the gray and brown curves do not change for $M = 25, 100, 900$ because Empirical DRO does not sample. The results show that DRO-BAS and BDRO perform better than Empirical KL and Wasserstein DRO when a sufficiently large number of samples is taken from the model ($M = 900$). However, for $M = 25$, Empirical DRO has better the OOS performance than model-based DRO approaches because more samples need to be taken to better approximate the model. For $M = 100$, Empirical DRO and the model-based methods are broadly comparable, with Empirical KL DRO having an advantage on the Truncated Normal DGP, likely due to misspecification of the likelihood.

### E.4. Cross-validation for choosing $\epsilon$

In practice, the decision-maker is unlikely to be able to calculate the optimal value of $\epsilon$ because they do not know the distributional form of the DGP. Instead, the decision-maker must calibrate $\epsilon$ from the available training data. One method for achieving this is to perform cross-validation (CV) by splitting the data into training folds, then evaluating the OOS performance of proposed solution on the test folds. We perform 10-fold CV on the Newsvendor for DRO-BAS with a univariate Normal DGP and $n = 100$ training observations. We train our model only on the training folds, then after solving for each candidate $\epsilon$, we calculate the CV mean and variance of the objective function *on the validation folds*.

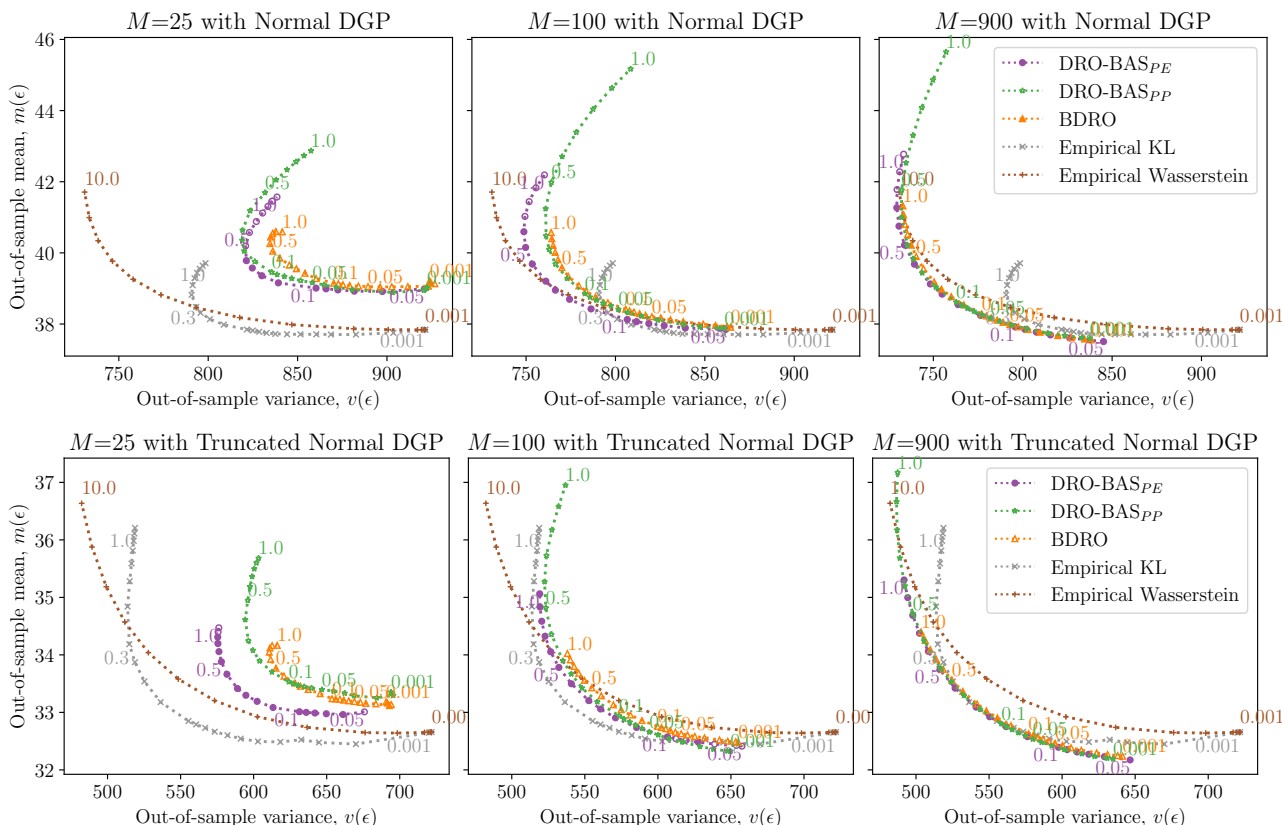

*Figure 8.* Comparison of the OOS mean-variance tradeoff on the Newsvendor for Empirical KL DRO and Empirical Wasserstein DRO vs Bayesian model-based DRO approaches on the Newsvendor Problem with Normal DGP (top) and Truncated Normal DGP (bottom).

The results in Figure 9 show how the decision-maker can choose a value of $\epsilon$ depending upon their appetite for risk. For example, if the decision-maker wishes to minimise the CV variance, then they should choose the value of $\epsilon$ marked by the stars in Figure 9. The triangle markers in the figure show a way to choose $\epsilon$ that compromises between the CV mean and variance by taking a average of the CV mean and the CV standard deviation. By considering the CV mean-variance tradeoff, the decision-maker is now well-informed to choose $\epsilon$ with knowledge of how the chosen value could perform on out-of-sample, unseen data.

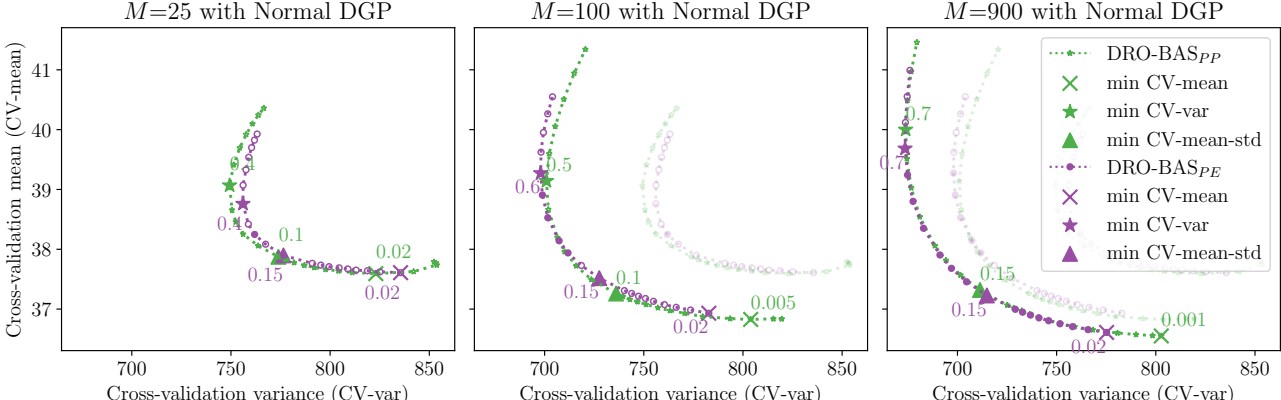

*Figure 9.* The *cross-validation* (CV) mean-variance tradeoff on the Newsvendor Problem with Normal DGP and $n = 100$ training observations. We highlight three ways the decision maker can choose $\epsilon$: firstly, minimise the CV-mean (cross markers); secondly, minimise the CV-var (star markers); and thirdly, minimise $\frac{1}{2}$ (CV-mean + CV-std) (triangle markers), where CV-std is the standard deviation of the objective function on the validation folds.

