# OpenReview forum: "Decision Making under the Exponential Family: Distributionally Robust Optimisation with Bayesian Ambiguity Sets"
_ICML.cc/2025/Conference — ICML 2025 spotlightposter_

### Official Review · Reviewer_GxWu · 2025-03-11

**Overall Recommendation:** 3

**Summary:**

The authors introduce a novel formulation for distributionally robust optimization based on Bayesian posterior updates. Their model leverages KL-divergence, and they propose two ambiguity set designs with efficient sampling methods. These methods are computationally better than Bayesian DRO under mild conditions. Experimental results demonstrate their computational and statistical benefits.

**Claims And Evidence:**

I had some questions related to expressions in Table 1:

1.	The authors do not clarify how the problem dimension $D$ influences the number of variables. It would be helpful to reference this explicitly, at least in Appendix.

2.	Regarding the column “Linear f”, it is nonclear why $BAS_{PE}$ and $BDRO$ exhibit a dependency on $D^2$. Given that the number of decision variables appears to be only $O(D)$ (and $O(M_{\theta} D)$) after the transformation in Line 290 or Equation (40), further clarification is needed.

**Essential References Not Discussed:**

I do not find the authors missing any important literature.

**Experimental Designs Or Analyses:**

I checked the validity of experimental designs and had several problems:

1.	I appreciate the visualization of Figures 2 and 3, particularly regarding the mean-variance frontier. It would be insightful to analyze which $\epsilon$ values would be practical for real-world implementations.

2.	In Figure 7 of Appendix, the authors present multiple $\epsilon$ values for each method across the whole time period. It would be beneficial to select the optimal $\epsilon$ via cross validation and compare their performance accordingly, as is common in DRO literature. Also, all different methods appear to collapse under $\epsilon = 1$, but this is not well visualized.

**Methods And Evaluation Criteria:**

The proposed method and evaluation criteria appear well-justified for the application.

**Other Comments Or Suggestions:**

1.	In the multi-product newsvendor, the objective formulation should be written as $h 1^{\top} max(0, x - \xi)$. The current formulation is of vector output.

2.	The texts in Figures 2 (especially) and 3 mix together across different methods. A clearer visualization strategy is to use an arrow to indicate the direction from 0.001 to 1 simply, instead of marking all the values there.

**Other Strengths And Weaknesses:**

Overall, the paper is well-written and clearly articulates its contributions. Below, I outline some concerns and areas for improvement:

1.	**Statistical Benefits**: Besides computational advantages, the statistical benefits of DRO-BAS are not entirely clear. Can authors further discuss how DRO-BAS compares to Bayesian DRO in terms of statistical performance? Additionally, under what conditions should one prefer DRO-BAS over Bayesian DRO given the prior? Based on the paper’s description, the observed experimental benefits seem primarily due to improved computational approximation.

2.	**Novel Considerations**: While I appreciate the authors’ interpretation for these Bayesian ambiguity sets, the model design in Equation (18) and Theorem 3.6 appears to closely align with the PDRO method from Iyengar et al. (2023) when using KL-divergence cases (from the standard KL reformulation technique from Hu and Hong (2013)). Both approaches define an ambiguity set centered at an estimated parametric distribution via KL-divergence. Although the interpretations differ, a discussion clarifying the novelty of the approach would be beneficial.

3.	**Scalability for General Models**: While the proposed approach performs well for exponential and normal distributions, posterior updates for general large-scale models may still be computationally expensive. This scalability challenge is a well-known limitation of Bayesian DRO.

**Questions For Authors:**

1.	Since the sampling number affects the performance of DRO-BAS, are there any Monte Carlo sampling guarantees similar to those in Iyengar et al. (2023) that provides strong performance guarantee?

2.	The choice of divergence functions appears flexible. For example, I do not see a fundamental issue when extending the design idea to $chi^2$-divergence. In such a case, Property 3.1 might only require the existence of a second moment. Could the authors elaborate on this or discuss necessary modifications? Such extensions would enhance  the practical applicability of the method, especially given concerns about KL-divergence’s conservativeness and little use in practice.

**Relation To Broader Scientific Literature:**

The paper contributes a novel approach by incorporating Bayesian posterior updates into the KL-DRO framework.

**Theoretical Claims:**

I reviewed the proofs for all the main results (except concrete examples in Appendix A) and did not find major issues that worth mentioning.

---

> ### Author Rebuttal · Authors · 2025-03-31
>
> Thank you for providing feedback that greatly improves the paper. Please see this anonymous link for Tables and Figures: https://github.com/ICML-anon-2025/paper-11717
>
> 1. **Table 1**: We have changed the table and caption to distinguish between variables and the input size of the dual (see link above). As you suggested, we will also add a brief discussion of Table 1 inside Sec. 3.5 with a more detailed explanation in the appendix.
>
> 2. **Practical choice of $\epsilon$ and cross-validation (CV)**:
> We have now implemented a 10-fold CV procedure for choosing $\epsilon$ from a set $E$ for BAS-PE and BAS-PP. As suggested, we took Fig. 7 for the Normal DGP Newsvendor. We train our model only on the training folds, then after solving for each $\epsilon \in E$, we visualise the CV mean and variance on the validation folds across all replications in Fig. 9 of the link provided above. On Fig. 9, we have marked the $\epsilon$ that achieves 1) the smallest CV mean, 2) the smallest CV variance, 3) the smallest CV mean/standard-deviation tradeoff.
>
> 3. **Statistical benefits of DRO-BAS over BDRO**:
> One key difference is that we have a single worst-case objective, leading to an interpretable closed-form worst-case distribution (Sec. 3.6), whereas BDRO does not (see response 2 to fJ7Y). At the limit of infinite observations, both DRO-BAS and BDRO converge to a KL-based ambiguity set based on the DGP. However, our analysis is straightforward and more intuitive compared to BDRO, which uses a much more complex argument (p.1286-1292) to prove where the expected worst-case concentrates.
> In terms of statistical properties, we follow suggestions from the DRO literature (Shapiro et al. 2023, Gotoh et al. 2021) and focus on the mean-variance frontier, which is a type of statistical trade-off, showcasing our domination over BDRO. See also response 2 to V1H1 (on the statistical behaviour of the solution).
>
> 4. **Interpretation of BAS in comparison to PDRO**:
> Although the dual formulations of BAS and PDRO with the KL are both model-based and look related, they are fundamentally different. Our ambiguity sets are a-posteriori informed, integrating prior beliefs and data evidence. Consequently, the resulting nominal distribution ($\mathbb{P}_n$ for BAS-PP and $\mathbb{P}\_{\hat{\eta}}$ for BAS-PE) contains all the information from our posterior beliefs, including their uncertainty quantification, unlike the point estimator approach of Iyengar et al., 2023. This allows us to propagate uncertainty from the posterior beliefs about the parameters to the ambiguity set. We will highlight this important distinction by expanding the lines 53-63 with the points above.
>
> 5. **Scalability**: If the posterior is not available in closed-form, then the sampling time will increase (similarly to all Bayesian DRO methods) unless one performs approximate inference such as Variational or Laplace approximations. In this paper, we provide efficient formulations for a big class of models - the exponential family (which goes beyond just the Normal and Exponential distributions). See also response 3 to oPX6. Moreover, working with the exponential family enables our tractable formulations of the DRO-BAS problem and leverages the conjugacy property, which greatly enhances the scalability of our method.
>
> 6. **Monte Carlo guarantees**: It would be valuable to theoretically examine the effect of the Monte Carlo sampling size to the out-of-sample cost, similar to our empirical analysis in Sec. 4. This is a non-trivial analysis as it requires establishing bounds on the worst-case KL objectives based on a nominal distribution and its empirical approximation. This is also likely the reason why Iyengar et al., 2023 only provide these types of guarantees for the Wasserstein and $\chi^2$-divergence, and why Shapiro et al., 2023 omit them for the KL-based BDRO. Such guarantees would be a notable contribution on their own as they would likely lead to similar results for other model and KL-based DRO methods (like PDRO and BDRO). We will include this as future work in Sec. 5.
>
> 7. **Choice of divergence**: Please see response 3 to fJ7Y for a discussion of other $\phi$-divergences and response 4 to oPX6 for the benefits of KL-based BAS. The advantages of the KL have also been widely discussed in the DRO literature (Hu et al. 2013, Shapiro et al., 2023) and the closely related DR Bayesian Optimisation literature (Husain et al., 2023). We see extending BAS to other distances or divergences as an interesting future research direction.
>
> 8. **Other comments**: We will correct the vector notation for $h$ in the newsvendor objective. Concerning visualisations in Fig. 2 & 3, we have kept the $\epsilon$ markers because we do not want to give a false impression that the line is continuous and the values of $\epsilon$ are equidistant, but we will work to make the figures more legible.
>
> [1] Husain H. et al. Distributionally robust Bayesian optimization with ϕ-divergences. 2023.

---

### Official Review · Reviewer_oPX6 · 2025-03-13

**Overall Recommendation:** 3

**Summary:**

This paper introduces KL-based DRO formulations with two kinds of Bayesian ambiguity sets, the posterior expectation and the posterior predictive. The authors show that both formulations can be recast into a direct minimization problem, with more efficient closed form worst-case risk solution for exponential family distributions. Empirical study also shows the effective of the proposed estimators.

## update after rebuttal

I hope the authors would add comparisons to BDRO, standard ambiguity set, $\phi$-divergence in revision, as promised by the authors in their rebuttal. But in the current form, I still lean towards accept weakly.

**Claims And Evidence:**

Yes.

**Essential References Not Discussed:**

Since the method is mainly based on KL-based DRO, the authors may want to discuss a seminal work of the more general $\chi^2$ divergence-based DRO.


[1] Duchi, J., & Namkoong, H. (2018). Variance-based Regularization with Convex Objectives. Journal of Machine Learning Research, 19, 1–55. http://jmlr.org/papers/v19/17-750.html.

**Experimental Designs Or Analyses:**

Yes. I have checked the portfolio experiments which are reasonable.

**Methods And Evaluation Criteria:**

Yes.

**Other Comments Or Suggestions:**

- Line 035, right column, the data **is** noisy.
- Line 290, Eq. 9,11,12, missin period.
- Line 201, Definition 3.4, function $h$ is not defined or remarked before being used.

The authors should proofread the symbols, commas and periods used in the paper, especially in the context around math equations.

**Other Strengths And Weaknesses:**

**Strengths**

It is novel to propose a Bayesian DRO formulation to tackle the worst-case risk instead of the expected worst-case risk. The examples with conjugate exponential family provide insights into applicability of the proposed methods.

**Weaknesses**

The derived upper bounds in the theoretical analysis look qualitative rather than quantitative. The adopted KL divergence is restrictive, whose efficient dual formulation is well-known thus makes most of the dual analysis look mathematically trivial. See my detailed comments below.

**Questions For Authors:**

The paper is well-written and the idea is novel. I have a couple of minor concerns though.

1. What's the point of the upper bound in Eq. 13? It resembles Eq. 8, but is not quantitative since it depends on $f$, $\theta$, which makes  the generalization property unclear. Is it possible to provide a more quantitative bound, for example, by adopting Rademacher complexities? Otherwise, the theoretical analysis in the paper except for the conjugate looks like trivial corollaries of results in Hu
& Hong, 2013.
2. The optimal radius selection in Section 3.4 and its approximation with empirical data makes sense. However, there is still a gap between posterior distribution and the true distribution. The choice in line 265, right column does not take into account such gap. In other words, given a confidence level, say $\tau$, and number of samples $M$, how can I choose the radius that yields a non-asymptotic bound for Eq. 21 to hold with probability at least $1 - \tau$?
3. The theoretical guarantees and computational efficiently are somewhat tightly coupled with exponential family distributions. The authors admit the limitations in Discussion as well. What properties should a family of distributions encompass to go beyond exponential family while retaining the advantages claimed in the paper?
4. Why do the authors choose KL divergence-based DRO instead of DRO based on Wasserstein distances or MMD?

**Relation To Broader Scientific Literature:**

The key contributions are related to the KL divergence-based distributionally robust optimization literature by approach it from a Bayesian perspective.

**Theoretical Claims:**

Yes. I have checked the proofs of Proposition 3.2 and 3.3, both of which look correct to me.

---

> ### Author Rebuttal · Authors · 2025-03-31
>
> Thank you for carefully considering our work and providing feedback that greatly improves the paper:
>
> 1. **Upper bound in Eq. 13**: There is potentially a misunderstanding here: our work does not rely on this bound and goes beyond this inequality to achieve the strong duality result (with equality) in Theorem 3.6. In detail:
>     - *"What's the point of the upper bound in Eq. 13?"*: It provides a weak duality result for general parametric Bayesian models without closed-form posteriors, e.g. Bayesian neural networks. The discussion after Proposition 3.3 explains the challenge associated with BAS-PE leading to the upper bound in Eq. 13. Exactly for this reason we study the widely used exponential family models in Sec. 3.3, allowing us to move from an upper bound (weak duality) to an equality (strong duality) in Theorem 3.6.
>     - *"It resembles Eq. 8, but is not quantitative since it depends on $f$ and $\theta$ which makes the generalization property unclear"*: The goal of the duality result is to reformulate the maximisation in Eq. 11 as a minimisation so that the objective can be jointly minimised in the Lagrange variable $\gamma$ and decision variable $x$. As a duality result, it necessarily relies on $f$ and $\theta$ as these are part of the worst-case risk (the same dependence is in Proposition 3.2, Theorem 3.6, and in the duality results of BDRO and other methods).
>     - *"Is it possible to provide a more quantitative bound, [...] Otherwise, the theoretical analysis in the paper, except for the conjugate, looks like trivial corollaries of results in Hu \& Hong, 2013."* The theoretical analysis in Sec. 3.3 is far from trivial and cannot be seen as corollaries of Hu et al. 2013 as their analysis is for divergences and not for expected divergences. The goal of Sec. 3.3 is to go beyond the weak duality bound in Eq. 13 and provide stronger results. In particular, it provides an important formulation of the expected KL divergence in Lemma 3.5, which is crucial to overcoming the difficulty of obtaining the convex conjugate of the expected KL. This leads to a strong duality result in Theorem 3.6 along with the results in Corollary 3.7, Eq. 20 and Sec. 3.6, which have not been derived before for an ambiguity set based on an expected divergence (rather than just a divergence). To avoid similar misunderstandings in the future, we will include the clarifications above before Proposition 3.3.
>
> 2. **Optimal radius selection**: Such a probabilistic guarantee would be very interesting but is challenging future work that we add to the discussion. It would require finite sample concentration results about the KL divergence with respect to the posterior distribution. This is challenging as evident from the lack of such results in the DRO/KLD literature. To provide an alternative method of selecting the tolerance level, we now also perform a cross-validation radius selection analysis (see response 2 to GxWu)
>
> 3. **Restriction to the exponential family**: As discussed in Sec. 5, this is an interesting topic for future work. The key property is that the family of distributions should have a closed-form expression for the expected KL divergence, such as we have in Lemma 3.5. Otherwise, one can apply Proposition 3.3 to obtain an upper bound on the worst-case risk. We will include this comment in the discussion at the beginning of Sec. 3.3. Thankfully, we have obtained such a result for the exponential family, which is widespread across probabilistic graphical models, as well as models in spatial statistics and time series analysis, which also rely heavily on the exponential family. Moreover, it plays a central role in Variational inference, where it is the de facto approximating family, making our work broadly applicable across Machine Learning domains.
>
> 4. **Choice of KL divergence (KLD) & missed paper in lit. review**: Firstly, the KLD is a natural choice as the Bayesian posterior itself targets the KLD minimiser between the model family and DGP. Hence, intuitively, the expected KLD and the KLD with respect to the posterior predictive become good measures of separation between our posterior beliefs and the DGP (see main text lines 119-127 and Shapiro et al. 2023). The KLD further allows us to obtain a strong dual formulation in Theorem 3.6 for the DRO-BAS-PE setting and tractable reformulations (see Sec. 3.5). Moreover, by using the KLD, DRO-BAS admits the formulation of the optimal radius and the worst-case distribution, which are very important for interpretability and applicability of the method, overcoming shortcomings of the KL-based BDRO. We will further discuss the challenges associated with extending the framework to other distances/divergences in future work (Sec. 5) (see response 3 to fJ7Y) and include the missed paper in our literature review.
>
> 5. **Typos and other comments**: We will correct the typos and properly define the function $h$, which corresponds to the scaling constant of the exponential family.

---

### Official Review · Reviewer_fJ7Y · 2025-03-13

**Overall Recommendation:** 3

**Summary:**

In this paper, the authors focus on Distributionally Robust Optimization and introduce two new ambiguity sets with the aim of informing the construction of ambiguity sets using Bayesian Statistics. More specifically, they use the posterior distribution to construct these ambiguity sets. In the first set, we consider all distributions whose distance from the posterior-predictive distribution is less than a parameter \(\epsilon\). In the second case, we take into account all distributions whose expected KL-divergence (w.r.t. the posterior distribution) from the nominal distribution is bounded by \(\epsilon\). For these two cases, the authors provide appropriate reformulations. They then demonstrate the performance of their approach on a newsvendor and a real-world portfolio optimization problem. They show that their approaches peform as good as the BDRO method or better while being easier to solve and taking less time.

## Update after Rebuttal
The authors have strengthened the paper with better numerical experiments. However, I decided to maintain my score after receiving the response of the authors.

**Claims And Evidence:**

The theoretical results provided in the paper appear sound. The numerical experiments demonstrating the benefit of this approach over the BDRO method appear sound, and both real and synthetic data are used well. The key limitation, I believe, is the limited comparisons to alternative methods. While BDRO is indeed the most comparable method, it would be interesting to see evaluations against most common DRO sets such as standard KL divergence sets and sets with other metrics to evaluate the advantages and disadvantages of the Bayesian approach.

**Essential References Not Discussed:**

None that I know of.

**Experimental Designs Or Analyses:**

The experimental analysis presented in the paper is sound and is similar to existing work in this domain.

**Methods And Evaluation Criteria:**

The authors demonstrate the performance of their method using a pareto frontier with the out-of-sample mean and variance of the solution. They use both real and synthetic data to demonstrate their approach. All this appears reasonable.

**Other Comments Or Suggestions:**

1. Have the authors also considered developing these sets for general phi-divergence metrics instead of just KL-divergence.

**Other Strengths And Weaknesses:**

Strengths:
1. Novel ambiguity sets which incorporate both data, prior beliefs and distance metrics
2. Tractable reformulations for these ambiguity sets
3. Discussion of how to choose tolerance parameters.
4. Numerical experiments with both synthetic and real data.

Weaknesses
1. While the comparison with other Bayesian DRO approaches is present, it is difficult to evaluate how the Baysian Ambiguity Sets perform when compared against other DRO approaches such as Wasserstein ambiguity sets etc.

**Questions For Authors:**

1. I suggest authors compare their approach against other standard ambiguity sets so we can better understand how useful the Bayesian Ambiguity Sets are in practice.
2. Can the authors discuss a bit about how the worst case distribution that is being identified by the two BAS and BDRO differ amongst each other.

**Relation To Broader Scientific Literature:**

The paper extends existing literature on the use of Bayesian Ambiguity Sets for distributionally robust optimization by introducing new types of ambiguity sets which combine together KL-divergence and posterior information from data. This allows us to simultaneously use data and prior beliefs.

**Theoretical Claims:**

I verified the key results associated with duality theory specifically Proposition 3.2, Proposition 3.3 and Theorem 3.6.

---

> ### Author Rebuttal · Authors · 2025-03-31
>
> Thank you for carefully considering our work and providing feedback and suggestions that greatly improve the paper:
>
> 1. **Additional comparisons to other DRO ambiguity sets**: Firstly, the suggested empirical-based ambiguity sets, based on the KL-divergence and Wasserstein distance, are fundamentally different to model-based approaches (e.g. DRO-BAS, BDRO) as they are fully data-driven and are not appropriate in cases where the decision-maker relies on a model to make a decision. There is already a discussion of the model-based vs empirical KL DRO setting in Appendix E.3, however, we will add a short statement in the introduction pointing the reader to this discussion.
> To this end, we have already included a comparison to the standard, empirical KL divergence ambiguity set that you suggested in Appendix E.3 and Figure 7. To further strengthen our empirical comparisons, and following your suggestion, we have also now conducted a comparison to the Wasserstein-based ambiguity set centred on the empirical measure for a well-specified (Normal DGP) and a misspecified (Truncated Normal DGP) Newsvendor example (see Figure 8 of this anonymous link: https://github.com/ICML-anon-2025/paper-11717). We observe that, for M sufficiently large, DRO-BAS outperforms all empirical methods. We will include this new comparison in Sec. 4 and in the Appendix.
>
> 2. **How do the worst case distributions identified by BAS and BDRO differ?**: Crucially, for BDRO, a single worst-case distribution does not exist because BDRO considers an expected worst-case approach (see Eq. 2, Figure 1 and the discussion on lines 67-79) rather than a worst-case approach, as advocated by DRO methods. In contrast, since both DRO-BAS formulations correspond to a worst-case risk minimisation objective, we obtain the worst-case distributions in Sec. 3.6. If we sample $\theta_j \sim \Pi$ from the posterior for BDRO, then one can obtain a worst-case distribution for each $\mathbb{P}\_{\theta_j}$ via the argument in Hu & Hong (2013), eq. (8). However, notice that in general, the minimiser of an expected objective is not the same as the average of the minimisers of the individual objectives as $\min\{f(x) + g(x)\} \geq \min\{f(x)\} + \min\{g(x)\}$ for any functions $f$ and $g$. Hence, even looking at the posterior mean or mode of the worst-case minimisers of the inner worst-case objectives in Eq. 2 would not necessarily give us a single distribution $p$ that yields a worst-case risk of the form $\mathbb{E}_{\xi \sim p} [f(x,\xi)]$ corresponding to the risk minimised by BDRO. In the limiting case of infinite data observations ($n \rightarrow \infty$) all of BDRO, BAS-PE and BAS-PP concentrate to a KL ambiguity set based on the data-generating process $\mathbb{P}^\star$ hence the worst case distribution is asymptotically the same for all methods but it is different (or non-existent for BDRO) for finite sample sizes.
> This further highlights a fundamental difference between BDRO, which opts for an expected worst-case objective, and DRO-BAS, which follows the common DRO route of defining a worst-case objective. DRO-BAS provides the decision-maker with an exact and interpretable formulation of the worst-case risk being minimised. We will expand further on these points in lines 60-63 where we discuss the main difference of our method to BDRO and also include the above points in Sec. 3.6, which deals with the DRO-BAS worst-case distribution.
>
> 3. **Extension to general phi-divergence metrics?**: This is something we are actively working towards and believe would be very valuable future work.
>     - The analysis for DRO-BAS-PP would involve the derivation of the dual formulation of the problem based on convex conjugate results for $\phi$-divergences. Results about the radius and the worst-case distribution might be more challenging and would rely on properties of the posterior predictive and $\phi$-divergences.
>     - The analysis for DRO-BAS-PE appears even more challenging: one would need to derive, in closed-form, an expression for the expected $\phi$-divergence under the posterior (similarly to Lemma 3.5). This would allow us to derive the convex conjugate form of the expected $\phi$-divergence, which is otherwise challenging to obtain. The convex conjugate expression of the expected divergence is essential in order to derive the strong duality result in Theorem 3.6, but the closed-form of the expected KL divergence further allowed us to obtain the results about the tolerance level selection (Sec. 3.4), a closed-form objective in the Gaussian case with linear cost function (Sec. 3.5) and the closed form of the worst-case distribution (Sec. 3.6).
>     - In the discussion, we will explain the challenges associated with extending BAS to other $\phi$-divergences and suggest this as important future work. We believe this framework opens the door to alternative BAS formulations grounded in expected distances or divergences, such as the $\phi$-divergence you mentioned.

---

> > ### Comment · Reviewer_fJ7Y · 2025-04-09
> >
> > Thank you for your responses and for answering my questions. After considering them, I have decided to maintain my score.

---

### Official Review · Reviewer_V1H1 · 2025-03-15

**Overall Recommendation:** 4

**Summary:**

The paper provides two new ways  to define the distributional robust counterparts for the  Distributionally Robust Optimization (DRO) taking into account posterior information, called Bayesian Ambiguity Sets (BAS) . In particular, the authors address the problem of the worst-case risk optimization where the wort-case distribution is taken subject to an ambiguity set. The first set is BAS with posterior predictive (BAS-PP), that is defined by bounding the KL divergence by $\epsilon$ from the parametric distribution defined by posterior expectation. The authors propose to use it in the cases when distributions have bounded Moment Generation Function, since the DRO has a closed form in this case. For the more general distributions, the authors propose to use the BAS based on posterior expectation (BAS-PE), which bounds the expectation of the LK divergence to the potential parametric distribution, where the expectation is taken over the posterior distribution of a parameter. The authors prove that this formulation allows DRO to have a closed form for the case of exponential family of distributions. The authors demonstrate that the DRO with their ambiguity sets allows one-stage dual problem formulation, whereas the closest benchmark Bayesian DRO by Shapiro et al. (2023) needs to define two-stage optimization and requires more sampling. They also provide an empirical comparison it terms of computational complexity of solving the DRO including the sampling time, and compare the mean-variance trade-offs of the solutions obtained by different DRO formulations.

## Update after rebuttal:

I thank the authors for the detailed response and I keep my original score.

**Claims And Evidence:**

Yes, as far as I can see. However, it would be good to add the direct references / hyperlinks to the proofs in the Appendix of each Lemma / Theorem directly after introducing them in the body of the paper.

**Essential References Not Discussed:**

Not that I'm aware of of

**Experimental Designs Or Analyses:**

The experiments seem to be quite thorough.

**Methods And Evaluation Criteria:**

Yes, it seems so.

**Other Comments Or Suggestions:**

There just a very few typos that I noticed, and comments I would like to make.

Typos:
- Line 268, Corollary 3.7: I think it should be $\mathcal A_{\epsilon}$ instead of $A_{\epsilon}$

Comments:
- Line 102 left side: Let x be a decision variable that minimizes a stochastic objective function... Maybe, that should be chosen to minimize? Otherwise it sounds like it is a solution already.
- Line 198, right side:  When defining exponential family with conjugate prior, could you please add the particular reference where it is taken from?
- Line 203 right side: Also, the function $h(\xi_i)$ in Definition 3.4 seems to be undefined. What $h$ stands for?
- Please add the references to the proofs in the Appendix directly after the corresponding results in the body of the paper.

**Other Strengths And Weaknesses:**

The paper is very clearly written, very clean, almost everything is properly defined. The first formulation DRO_BAS_PP was proposed before in the non-bayesian framework, but extended here to the Bayesian framework, as they discuss in the paper. The proposed second formulation DRO_BAS_PE seems to be novel. The result of the closed form formulation of the DRO in this case for exponential family of distributions also seem to be novel.

**Questions For Authors:**

- Line 248 right side: It follows that ...(21) - Where does it follow from? Here the flow was not very clear to me. Please clarify.
- Figure 2. Can you please explain why for increasing $\epsilon$ until 1, in some of the Pareto-curves the out-of-sample variance $v(\epsilon)$ grows again?

**Relation To Broader Scientific Literature:**

The paper provides a new way to define the ambiguity sets for the distributional robust optimization that take into account the bayesian posterior, and allow simpler optimization formulations for certain distribution families than the previous formulations.

**Theoretical Claims:**

I tried to follow the derivations and claims in the body of the paper, and they all seem to be correct. I did not check the appendix.

---

> ### Author Rebuttal · Authors · 2025-03-31
>
> Thank you for carefully considering our work and providing feedback that greatly improve the paper:
> 1. **Clarification of line 248 right side**:  $\epsilon^{\star}\_{\text{PE}}(n)$ in Eq. 20 is defined as the expected KL divergence between the data-generating process $\mathbb{P}^\star$ and model $\mathbb{P}\_\eta$, indexed by the natural parameter $\eta$. For any $\epsilon \geq \epsilon^\star\_{\text{PE}}(n)$ we have that $\mathbb{E}\_{\eta \sim \Pi}[d\_{KL}(\mathbb{P}^\star || \mathbb{P}\_\eta)] = \epsilon^\star\_{\text{PE}} \leq \epsilon$ and hence $\mathbb{P}^\star \in \mathcal{A}\_{\epsilon}(\Pi)$ by definition of the BAS-PE ambiguity set $\mathcal{A}\_\epsilon(\Pi)$. Since $\mathbb{P}^\star$ belongs to the ambiguity set, the worst-case risk will be at least as big as the risk under $\mathbb{P}^\star$ (since the supremum over a set is always at least as big as the value attained by any of its elements), resulting in Eq. 22. We will add this explanation to the main text and highlight its importance: if one chooses the radius at least as large as $\epsilon^\star\_{\text{PE}}(n)$ then the worst-case risk minimised by DRO-BAS-PE will upper bound the true risk under $\mathbb{P}^\star$.
> 2. **Explanation of variance behaviour for increasing values of $\epsilon$ in Fig. 2**: The reason is well-understood but quite challenging to convey compactly: the behaviour of the OOS variance $v(\epsilon)$ in the Newsvendor experiments is due to the behaviour of the variance of the solution (denoted by $v\_{x}(\epsilon)$) (see Fig. 10 in this anonymised link: https://github.com/ICML-anon-2025/paper-11717) across epsilons. In turn, the variance of the solution is driven by the Newsvendor asymmetric cost function and its interplay with the worst-case distribution for each $\epsilon$. The Newsvendor objective is piece-wise linear with 2 pieces, and hence, the true risk (expected cost under the DGP, in this example a Normal) has a similar two-piece behaviour. The optimal solution lies at the intersection of the two pieces of the true risk. The piece corresponding to larger solutions has a significantly smaller slope (see 4th plot of Fig. 10), leading to a smaller true risk.
>     - Let’s consider a fixed value of $M=25$. For small values of $\epsilon$, the variance of the solution $v\_{x}(\epsilon)$ is smaller because fewer distributions are included in the ambiguity set; hence, the obtained solution is fairly stable over replications (see first plot of Fig. 10). However, because BAS likely does not include the DGP, and the decision-making is risk-prone, the OOS variance of the cost function $v(\epsilon)$ is large (see the Pareto curve of the first plot of Fig. 4).  In this regime, the OOS variance reduces as we increase epsilon and better capture the DGP.
>     - As $\epsilon$ increases further, the ambiguity sets start including a lot more distributions, leading to the mean solution moving to values larger than the optimal solution where the slope of the true risk is smaller (see 4th plot of Fig. 10). This makes sense as, as we increase $\epsilon$, we become more conservative. However, as $\epsilon$ increases to very big values (in this case $> 0.5$), the ambiguity set contains a lot of arbitrary distributions, significantly increasing the OOS cost. The methods then push the solution towards smaller values of $x$ as can be observed by the big variance $v\_{x}(\epsilon)$ on the first plot for larger values of $\epsilon$.
>     - This behaviour is common for all values of $M$, however, there is an important distinction: for smaller values of $M$ (25, 100), the error bars extend way below and above the optimal solution. If we associate this with the form of the true risk on the last plot, we can expect a very high variance $v(\epsilon)$ of the OOS cost. On the other hand, the empirical variance of the solution $v\_{x}(\epsilon)$ for large M ($M = 900$) does not extend below the optimal solution by a high degree meaning that as the optimisation becomes more exact, the methods suggest staying on values bigger than the optimal solution which correspond to the smaller slope of the true risk. This is why in the third plot of Fig. 4 in the main text ($M = 900$), the OOS variance $v(\epsilon)$ does not increase by a lot for big values of $\epsilon$.
>     - We believe this plot and the above explanation give great intuition on how the specific form of the cost function can affect the behaviour of the method and will hence include it in the Appendix.
>
> 3. **Other comments**: We will change the statement in line 102 to your suggestion. We will further add a reference to Definition 3.4, noting that we are following the notation of Murphy et al. 2023 for the conjugate exponential family. We accidentally omitted the definition of $h(\xi)$, which is referred to as the scaling constant.
>
> 4. **Typos and references to proofs**: We will correct the typo and add direct references to the location of each proof after each mathematical statement.

---

### Decision · Program_Chairs · 2025-05-01

**Decision:**

Accept (spotlight poster)

**Comment:**

All reviewers agree that this well-written paper presents solid contributions. The paper is clear, well-motivated, and organized well. The theoretical results are novel and significant, and they are explained well. The numerical contributions are insightful too. The reviewers did not identify any significant concerns.